# Research Models to Study Ferroptosis’s Impact in Neurodegenerative Diseases

**DOI:** 10.3390/pharmaceutics15051369

**Published:** 2023-04-29

**Authors:** Inês Costa, Daniel José Barbosa, Vera Silva, Sofia Benfeito, Fernanda Borges, Fernando Remião, Renata Silva

**Affiliations:** 1Associate Laboratory i4HB—Institute for Health and Bioeconomy, Faculty of Pharmacy, University of Porto, 4050-313 Porto, Portugal; inessilvacosta@hotmail.com (I.C.); veralssilva17@gmail.com (V.S.); remiao@ff.up.pt (F.R.); 2UCIBIO—Applied Molecular Biosciences Unit, Laboratory of Toxicology, Department of Biological Sciences, Faculty of Pharmacy, University of Porto, 4050-313 Porto, Portugal; 3TOXRUN—Toxicology Research Unit, University Institute of Health Sciences, CESPU, CRL, 4585-116 Gandra, Portugal; 4Instituto de Investigação e Inovação em Saúde (i3S), Universidade do Porto, 4200-135 Porto, Portugal; 5CIQUP-IMS—Department of Chemistry and Biochemistry, Faculty of Sciences, University of Porto, 4169-007 Porto, Portugal; ester.benfeito@fc.up.pt (S.B.); fborges@fc.up.pt (F.B.)

**Keywords:** ferroptosis, in vitro models, in vivo models, neurodegenerative diseases, neurodegeneration

## Abstract

Ferroptosis is a type of regulated cell death promoted by the appearance of oxidative perturbations in the intracellular microenvironment constitutively controlled by glutathione peroxidase 4 (GPX4). It is characterized by increased production of reactive oxygen species, intracellular iron accumulation, lipid peroxidation, inhibition of system Xc-, glutathione depletion, and decreased GPX4 activity. Several pieces of evidence support the involvement of ferroptosis in distinct neurodegenerative diseases. In vitro and in vivo models allow a reliable transition to clinical studies. Several in vitro models, including differentiated SH-SY5Y and PC12 cells, among others, have been used to investigate the pathophysiological mechanisms of distinct neurodegenerative diseases, including ferroptosis. In addition, they can be useful in the development of potential ferroptosis inhibitors that can be used as disease-modifying drugs for the treatment of such diseases. On the other hand, in vivo models based on the manipulation of rodents and invertebrate animals, such as *Drosophila melanogaster*, *Caenorhabditis elegans*, and zebrafish, have been increasingly used for research in neurodegeneration. This work provides an up-to-date review of the main in vitro and in vivo models that can be used to evaluate ferroptosis in the most prevalent neurodegenerative diseases, and to explore potential new drug targets and novel drug candidates for effective disease-modifying therapies.

## 1. Neurodegenerative Diseases—An Overview

Neurodegenerative diseases (NDs) are debilitant conditions affecting the nervous system, being a common and rising cause of morbidity and mortality worldwide, particularly in the elderly [1]. These diseases are characterized by the selective dysfunction and progressive and gradual loss of synapses and neurons. They may involve the appearance of pathologically altered proteins, which primarily deposit in the human brain and spinal cord. These diseases show a wide spectrum of clinical presentations [2,3] and, generically, can be classified into two large groups. The first group concentrates the pathologies where cognitive deficits and memory loss or dementia occur, and includes Alzheimer’s disease, frontotemporal dementia, mixed dementia, dementia with Lewy bodies, and vascular dementia. The second group includes pathologies that mainly affect the locomotor system, such as Parkinson’s and Huntington’s diseases, amyotrophic lateral sclerosis, multiple sclerosis, and spinocerebellar ataxias [1,4]. These diseases present a complex and multifactorial nature, with several pathophysiologic mechanisms being already identified, many of them common to such diseases and which contribute to disease development and progression (Figure 1).

Alzheimer’s disease (AD) arises from neuronal loss (cortical and hippocampal cell loss) at a specific time in a patient’s life, thus being associated with aging. Approximately 60% to 80% of dementia cases correspond to AD [5], which represents the most common neurodegenerative disease. The symptoms experienced by patients with this pathology include persistent and frequent memory difficulties, vague speech, delay in performing routine activities, emotional unpredictability, and inability to understand questions and instructions [6]. The main markers of this pathology include the extracellular deposition of amyloid-β (Aβ) peptide in senile plaques, the intraneuronal accumulation of hyperphosphorylated Tau protein (leading to the formation of intracellular neurofibrillary tangles—NFTs), brain atrophy, neuronal and synaptic loss, and chronic inflammation [7]. Other mechanisms are also involved in the pathophysiology of AD, such as mitochondrial dysfunction, iron overload, and oxidative stress [8].

Parkinson’s disease (PD) is a neurological disorder that involves the progressive impairment of voluntary motor control. It affects about 0.1–0.2% of the general population and 1% of the population over 60 years, being the second most common neurodegenerative disease [9,10]. It is characterized by the degeneration or loss of dopaminergic neurons in the *substantia nigra* (SN) *pars compacta* and by the formation of Lewy bodies [cytoplasmatic aggregations of α-synuclein, (α-syn) in dopaminergic neurons]. Furthermore, multiple mechanisms have also been strongly connected to the neuronal loss observed in PD, such as mitochondrial dysfunction, iron accumulation, oxidative stress, neuroinflammation, among others [11], and which are features common to other NDs.

Amyotrophic lateral sclerosis (ALS) is a devastating and fast-progressing NDs characterized by the selective dysfunction and loss of motor neurons in specific brain regions, including the brain stem, motor cortex, and spinal cord, consequently leading to paralysis and death [12]. The incidence of this disease is approximately 1.2–6 cases per 100,000 persons annually [13]. Even though the pathophysiological mechanisms underlying ALS are not yet totally clarified, the aggregation and accumulation of ubiquitinated proteinaceous inclusions in the motor neurons are considered a neuropathological hallmark of the disease [14]. In addition, other mechanisms, including oxidative stress, excitotoxicity, defects in axonal transport, neuroinflammation, and mitochondrial dysfunction are also involved in ALS pathophysiology. Furthermore, approximately 90% of all cases of ALS occur sporadically (no family history associated), while 10% of ALS cases are familial and usually associated with dominantly inherited autosomal mutations that occur in distinct genes, including superoxide dismutase 1 (*SOD1*), *C9orf72,* and TAR DNA-binding protein 43 (*TDP*-43) genes [15].

Huntington’s disease (HD) is a rare autosomal dominant neurodegenerative disorder affecting 2.71 per 100,000 persons worldwide [16]. HD is caused by a polymorphic sequence of three CAG nucleotides in the first exon on the *Huntingtin* gene (*HTT*) that encodes for an expanded polyglutamine stretch in the Htt protein [17]. The main trigger factor of HD is the cleavage and aggregation of the mutant Htt (mHtt) into toxic macromolecules that consequently cause neuronal degeneration and cell death. These fragments show distinct toxicity properties, depending on where they are located. While in the cytoplasm, these fragments may disrupt the systems responsible for processing abnormal proteins, including chaperones, autophagy, and proteasome in neurons, when the toxic macromolecules permeate the nucleus of neuronal cells they can potentially interfere with the transcription of antioxidant genes [18]. However, the localization of toxic fragments both in the nucleus and in the cytoplasm promotes several mitochondrial alterations, including a decrease in adenosine triphosphate (ATP) levels and an increased production of reactive oxygen species (ROS), leading to oxidative stress [19].

Multiple sclerosis (MS) is recognized as a chronic inflammatory and demyelinating disease characterized by axonal degeneration and neuronal loss. The prevalence varies depending on the area of the globe, with high levels being detected in North America and Europe (more than 100/100,000 persons) and lower levels in Eastern Asia and sub-Saharan Africa (2/100,000 persons) [20]. The myelin damage characteristic of the disease is associated with an impaired ability of oligodendrocytes to regenerate, as shown in the brain of MS patients [21]. Overall, this leads to axonal degeneration, which represents the main cellular mechanism underlying the permanent disability consequent to irreversible neuronal damage [22]. However, other mechanisms, including, iron hyperaccumulation, oxidative stress, and neuroinflammation have also been reported in MS.

Friedreich’s ataxia (FRDA) is recognized as an autosomal recessive disorder originating from reduced levels of frataxin (FXN, a mitochondrial iron chaperone protein), consequent to a large GAA triplet-repeat expansion within the first intron of the frataxin-encoding gene. It is characterized by mitochondrial dysfunction, iron overload, oxidative stress, and neuroinflammation. In Europe, FRDA is the most common inherited ataxia with a prevalence of 1/20,000 in south-west Europe and 1/250,000 in northern and eastern Europe [23]. This disease is characterized by progressive gait and limb ataxia with associated extensor plantar responses, limb muscle weakness, decreased vibratory sense, dysarthria, proprioception, and absent lower limb reflexes [24,25].

Apart from the previously mentioned medical conditions, many NDs are well-identified and described in the literature. However, due to their lower prevalence, this review will mainly focus on the previously described diseases.

## 2. Neurodegeneration and Ferroptosis

Although the etiology of many NDs remains largely unknown, it is generally accepted that these diseases present common molecular and cellular mechanisms contributing to their pathogenesis [1]. These pathophysiological mechanisms include oxidative stress, abnormal protein misfolding and aggregation, inflammation, mitochondrial dysfunction, dysregulation of both calcium or metals’ homeostasis, excitotoxicity, and ferroptosis, among others [3,26,27,28,29].

Oxidative stress that occurs as a result of an imbalance between oxidants and antioxidants in a specific cellular, tissue, or organ system for a period of time represents a process that involves the disruption of redox signaling pathways [30,31]. When this process occurs, cells counteract the oxidant effects and try to restore the redox balance [32,33]. However, increased production of reactive species damages the integrity of several biomolecules, including lipids, proteins, and DNA/RNA, and contributes to cell degeneration which, in the brain, may lead to the degeneration of neuronal cells [34]. Since the brain consumes a large amount of oxygen, it is a particular target for oxidative stress. ROS production is initiated by the reduction of molecular oxygen (O_2_) to the superoxide radical (O_2_^•−^) by electron uptake. The superoxide radical can be further converted to hydrogen peroxide (H_2_O_2_), in a reaction catalyzed by superoxide dismutase (SOD), leading to the formation of the highly reactive hydroxyl radical (^•^OH) through the Fenton reaction catalyzed by ferrous iron (Fe^2+^) [35,36,37].

To prevent the cellular damage induced by oxidant radicals, cells are equipped with a very efficient antioxidant system, composed of antioxidant enzymes [SOD, catalase (CAT), glutathione reductase (GR), glutathione peroxidase (GPx), thioredoxin reductase (Trx), and glutaredoxine] and other non-enzymatic molecules [reduced glutathione (GSH), vitamins A, E, and C, Coenzyme Q10, and melatonin]. Overall, the antioxidant system components have the capacity to reduce distinct chemical structures, maintaining a balance between pro- and antioxidant agents, thus repressing oxidative stress [38].

Mitochondria is the main cellular source of ROS, producing 1–5% in normal physiological conditions. Complexes I, II, and III produce ROS in the mitochondrial matrix. At the mitochondrial complexes I and II, the electrons released from nicotinamide adenine dinucleotide (NADH) or flavin adenine dinucleotide (FADH_2_) are captured by oxygen, leading to the production of O_2_^•−^. These processes can also occur in the cytochrome P450-dependent oxygenases and nicotinamide adenine dinucleotide phosphate-oxidase (NOX) located on the cell membrane due to the flux of electrons between complexes III and IV [39].

Neuroinflammation, an inflammatory process that involves both the innate and adaptative immune system in the central nervous system (CNS), contributes to the impairment of normal brain function, triggers neuropathological events, and is observed in several NDs. Microglia are the main contributor to neuroinflammation. In the normal healthy brain, microglia can be activated by many substances, acquiring different functions like scavenging for neuronal debris, promotion of neuroplasticity, and maintenance of healthy brain physiology, among others [40]. When the CNS homeostasis is disturbed, microglia are activated and secrete inflammatory mediators (cytokines, chemokines, glutamate, and prostaglandins), which will activate astrocytes to induce a secondary inflammatory response [41].

A new type of programmed cell death, called ferroptosis, was identified by Dixon in 2012 [42] and has been increasingly recognized in the pathophysiology of distinct NDs [27,43]. In 2018, the Nomenclature Committee on Cell Death (NCCD) defined ferroptosis as “a form of regulated cell death (RCD) initiated by oxidative perturbations of the intracellular microenvironment that is under constitutive control of Glutathione peroxidase 4 (GPX4) and which can be inhibited by iron chelators and lipophilic antioxidants” [44].

Mechanistically, ferroptosis is characterized by increased iron accumulation, ROS overproduction (through the Fenton reaction) and lipid peroxidation, inhibition of system Xc-, GSH depletion, and decreased GPX4 activity (Figure 2) [45]. The accumulation of lipid peroxides [mainly phosphatidylethanolamine-OOH (PE-OOH)] and the action of iron as a catalyst regulator seem to be the two major contributors for ferroptosis initiation. In addition, the generation of ROS from the iron-catalyzed Fenton reaction and the decrease in antioxidant defenses (particularly GSH) can also contribute to the initiation of ferroptosis [46].

Lipid peroxidation is a key process in ferroptosis. Polyunsaturated fatty acids (PUFAs), given their structure [particularly arachidonic acid (AA) and adrenic acid (AdA)], represent the preferential substrates for lipid peroxidation. It can be enzymatically catalyzed, involving the enzymes Acyl-CoA Synthetase Long Chain Family Member 4 (ACSL4), lipoxygenase (LOX), and Lysophosphatidylcholine Acyltransferase 3 (LPCAT3) (Figure 2) [27,47]. Alternatively, lipid peroxidation may also be prompted by a nonenzymatic pathway, which is mainly dependent on Fe^2+^ to catalyze the decomposition of H_2_O_2_ and L–OOH, through the Fenton reaction, producing ^•^OH and lipid alkoxy groups. This promotes the accumulation of ROS, which leads to lipid peroxidation and, ultimately, to ferroptosis [48].

Iron is a metal with several physiological activities in the human body. It is very important for brain function, as it participates in the biosynthesis of neurotransmitters, in mitochondrial respiration, in myelin synthesis, among other fundamental functions. However, when in excess, this metal presents deleterious effects on the cells. A likely consequence of iron overload is an exaggerated generation of ROS via the Fenton reaction, lipid peroxidation, and ferroptosis [48]. Deferoxamine and deferiprone, iron-chelating agents that bind free iron in a stable complex, have ferroptosis inhibitory activity by removing the excess of iron [42,49,50].

On the other hand, another important mechanism correlated with ferroptosis is the inhibition of system Xc-. This system is an amino acid anti-transporter located in the plasma membrane, which exchanges intracellular glutamate and extracellular cystine at a ratio of 1:1 (Figure 2) [42]. Intracellularly, the imported cystine is then reduced to cysteine, which is needed for the biosynthesis of GSH. Therefore, the system Xc- transporter has a high impact on the biosynthesis of GSH [45]. Erastin, sulfasalazine, sorafenib, and glutamate are known inhibitors of system Xc-, leading to ferroptosis (ferroptosis inducers) [51,52,53,54,55,56]. Deferoxamine (DFO) activates the anti-transporter activity, stimulating the biosynthesis of GSH, behaving, therefore, as a ferroptosis inhibitor [57]. The GPX4 enzyme reduces toxic lipid hydroperoxides (L–OOH) into the corresponding alcohols (L–OH), while converting GSH into glutathione disulphide (GSSG, oxidized glutathione). Thus, since the activity of system Xc- limits the biosynthesis of GSH, GPX4 activity is indirectly dependent on it. Accordingly, the inhibition of this enzyme prompts the accumulation of lipid peroxides, which have the capacity of damaging the lipid bilayer of the plasma membrane, in favor of ferroptosis [45]. RSL3, withaferin, and FIN56 (ferroptosis inducers) inhibit the activity of GPX4, preventing the conversion of L–OOH to L–OH, and consequently, promoting lipid peroxidation and ferroptosis [58,59,60,61,62,63]. FINO2 indirectly inhibits GPX4 activity and oxidizes ferrous iron, thus causing widespread lipid peroxidation. On the other side, Ferrostatin-1 (Fer-1), Deferoxamine, and Liproxsatin-1 (Lip-1) promote GPX4 activity, behaving as ferroptosis inhibitors [64,65].

Other processes and regulators, including nicotinamide adenine dinucleotide phosphate (NADPH), the transcription factor nuclear factor erythroid 2-related factor 2 (Nrf2), and mitochondrial activity are also involved in ferroptosis [47,66,67,68,69,70,71,72].

Ferroptosis has been increasingly recognized in the pathophysiology of several NDs [43]. AD is characterized by increased iron accumulation in the brain (iron overload colocalizing with neuronal degeneration and brain atrophy), which potentiates Aβ deposition and Tau hyperphosphorylation and aggregation, increased levels of ROS and lipid peroxides (co-localized with amyloid plaques), and decreased levels of GSH and GPX4, disrupting redox homeostasis [43,73]. Similarly, depletion of GSH and GPX4 (GPX4 is associated with neuromelanin production in the SN and dystrophic axons in the PD brain) [74], lipid peroxidation [75,76], system Xc- deregulation [77], and iron accumulation (iron is correlated with an increased risk of α-syn fibers’ formation) [78] have been reported in distinct in vitro and in vivo models of PD.

ALS is also characterized by iron accumulation (iron accumulation was detected in the spinal cord of SOD1G37R transgenic mice) [79], lipid peroxidation [levels of 4-hydroxy-2-nonenal (4-HNE) in the serum and cerebrospinal fluid were significantly elevated in sporadic ALS patients] [80], and decreased GPX4 levels (ablation of GPX4 in neurons of a model of ALS resulted in rapid paralysis and severe muscle atrophy) [81]. Likewise, an accentuated increase of 4-HNE adducts in the *caudate nucleus* and *putamen* of the human HD brain [82] indicated abnormal lipid peroxidation. In addition, decreased intracellular GSH levels [83], as well as dysregulation of iron levels [84], have also been demonstrated in HD patients.

In MS, several studies have reported reduced levels of GSH and GPX4 [85], dysregulation of the system Xc-, and increased lipid peroxidation [86], clearly implicating ferroptosis in disease pathogenesis. Finally, FRDA patients [87] and mice models of the disease [88] show a decrease in GSH levels and in the expression of the gene that encodes for system Xc-. Furthermore, increased levels of lipid peroxidation markers were also observed [89], providing additional support for the involvement of ferroptosis in FRDA.

Overall, these data support a role for ferroptosis in the pathogenesis of distinct and unrelated NDs, as many of them show iron hyperaccumulation within the brain, lipid peroxidation, oxidative stress, and depletion of GSH. In fact, the nervous system relies heavily on iron for energy generation and brain cells are highly susceptible to oxidative stress. The antioxidant enzyme GPX4 is markedly expressed in neurons and glial cells and represents a powerful tool to counteract oxidative stress [90]. However, the brain is one of the most susceptible and sensitive organs to lipid peroxidation, due to the high oxygen consumption rate and the elevated amount of lipids [91]. All these factors render cells susceptible to degeneration and death characteristic of NDs. Nevertheless, it remains unclear how ferroptosis contributes to the clinical manifestation of these pathologies, including their onset and progression. Therefore, for a clear elucidation of ferroptosis’s involvement in the onset and progression of NDs, it is imperative to understand the development and validation of relevant in vitro and in vivo models. Understanding how the ferroptosis cascade is activated in such diseases, and the mechanisms involved, would help in defining strategies to overcome the unwanted effects of ferroptosis in the human body, and, particularly, will prompt the development of new disease-modifying drugs for the treatment of NDs.

## 3. Ferroptosis Inducers and Inhibitors

With the progress made in recent years towards understanding the underlying mechanisms of ferroptosis, the study of compounds with the ability to induce and inhibit this process of cell death has become extremely important. The main ferroptosis inducers and inhibitors are represented in Figure 3, as well as their targets.

Ferroptosis inducers can be classified according to their corresponding mechanism of action. Compounds like erastin, sorafenib, sulfasalazine, and glutamate act by inhibiting system Xc-, thus preventing the import of cystine, consequently reducing GSH levels. Additionally, RSL3, statins, Ml162 and Ml210, FIN56, FINO2, and withaferin inhibit GPX4 activity.

If ferroptosis induction can be beneficial in some pathologies, such as cancer, in other diseases, including NDs, stroke, and kidney diseases, its inhibition represents the desired strategy. Ferrostatin-1, liproxstatin-1, a-tocopherol, N-acetylcysteine, Zileuton, CoQ10, deferoxamine, deferiprone, FSP1, and BH4 are classified as ferroptosis inhibitors. Their mechanisms of action include blockage of lipid peroxidation, attenuation of ROS production, iron chelation, stimulation of GSH biosynthesis, and GPX4 activation. For a comprehensive review of ferroptosis induction and inhibition, and the involvement of this type of cell death in brain diseases, see [43].

## 4. Models to Study Ferroptosis in Neurodegeneration

The use of in vitro and in vivo research models in neuroscience is essential to gain insights into the mechanisms underlying pathological processes. In addition, they could have a fundamental role in the discovery and screening of novel compounds with potential therapeutic applications.

In fact, several in vitro and in vivo approaches have been already developed and employed to deeply study the etiology and pathogenesis of a wide range of NDs and the involvement of ferroptosis in such conditions [92]. In the following sections, the most suitable in vitro and in vivo models that could be successfully applied in the study of ferroptosis in NDs are highlighted.

### 4.1. In Vitro Models

#### 4.1.1. SH-SY5Y Cell Line

One of the most commonly used in vitro models applied in the study of NDs is the SH-SY5Y cell line [92]. This neuroblastoma cell line is a cloned subline of the neuroblastoma SK-N-SH cell line, which was originally established from a bone marrow biopsy of a neuroblastoma patient [93]. The neuronal phenotype of the SH-SY5Y cells can be modulated by using different protocols of terminal differentiation [94].

The neuronal differentiation of SH-SY5Y cells involves a series of specific processes, which include the formation and extension of neuritic processes, the increase in the electrical excitability of the plasma membrane, the formation of functional synapses, as well as the synthesis of neurotransmitters, and the expression of neurotransmitter receptors and neuron-specific enzymes [95,96]. When undifferentiated, SH-SY5Y cells are morphologically characterized by a neuroblast-like phenotype with non-polarized cell bodies. However, following treatment with specific differentiating agents, SH-SY5Y cells become morphologically similar to primary neurons [97]. The most commonly used method for SH-SY5Y cell differentiation uses retinoic acid (RA), at concentrations ranging from 5 μM to 100 μM and for a period of time spanning from 24 h to 21 days [94]. Retinoic acid is a derivative of vitamin A, which inhibits cellular growth and promotes cellular differentiation [97]. It also upregulates the expression of neuronal and dopaminergic (DAergic) markers and enhances cellular susceptibility to DAergic neurotoxins [96]. Another method for SH-SY5Y cell differentiation relies on the sequential treatment with RA and 12-O-Tetradecanoylphorbol-13-acetate (TPA). This protocol differentiates SH-SY5Y to a more characteristic DAergic phenotype, when compared to RA differentiation alone. A third approach for SH-SY5Y cell differentiation involves the sequential treatment with RA and brain-derived neurotrophic factor (BDNF). This differentiation protocol decreases cell proliferation and produces a homogeneous cellular population expressing neuronal markers [abundant neuritic processes, microtubule-associated protein-2 (MAP2), growth-associated protein-43 (GAP-43), and neuron-specific enolase (NSE)] [98]. The main advantages and limitations of this cell line are represented in Figure 4 and listed in Appendix A.

In PD studies, SH-SY5Y cells are widely used as an in vitro model to mimic impaired dopamine homeostasis, and to elucidate the mechanism(s) underlying 1-Methyl-4-phenylpyridinium (MPP^+^)-induced neurotoxicity, as these cells express both dopamine receptors and the dopamine transporter (DAT) [MPP^+^ is the toxic metabolite of 1-methyl-4-phenyl-1,2,3,6-tetrahydropyridine (MPTP) and a neurotoxin used to induce a PD phenotype] [99].

To investigate the involvement of ferroptosis in PD, Ito et al. explored the MPP^+^-induced cell death of SH-SY5Y cells [cells were previously differentiated with retinoic acid (10 μM for 7 days) and BDNF (50 ng/mL) for a further 5–6 days]. They demonstrated that MPP^+^ induced the non-apoptotic cell death of differentiated SH-SY5Y cells (5 mM for 48 h) as the cell death was not blocked upon incubation with two pancaspase inhibitors, Z-VAD-FMK and QVD-OPh. Furthermore, although MPP^+^-mediated cell death was blocked by the specific necroptosis inhibitor necrostatin-1, this effect was independent of receptor-interacting serine/threonine-protein kinase 1/3 (RIP1/RIP3). This suggests that MPP^+^-induced cell death was independent of necroptosis and indicates that RIP1 (necrostatin-1 mechanism of action involves the direct blockage of RIP1) was not the target molecule of necrostatin-1. To find a correlation between MPP^+^-cell death and ferroptosis, the group also analyzed the capability of ferroptosis inhibitors to counteract MPP^+^-induced cell death. Accordingly, MPP^+^-mediated cell death was also strongly inhibited by ferrostatin-1 (1–5 μM) and DFO (100–500 μM), known ferroptosis inhibitors, and by Trolox (an antioxidant hydrophilic analogue of vitamin E, 100–500 μM). The MPP^+^-induced death was accompanied by the accumulation of oxidized lipids, an effect strongly inhibited by ferroptosis inhibitors. This suggests a potential relationship between MPP^+^-induced cell death and ferroptosis (especially the involvement of lipid ROS). However, in contrary to ferroptosis (where p53 was found to increase cells’ sensitivity to ferroptosis by downregulating *SLC7A11* expression, the gene coding for system Xc-; as well as small mitochondria and no ATP depletion were detected [42]), MPP^+^-mediated cell death was independent of p53 and involved ATP depletion and mitochondrial swelling. Furthermore, contrarily to MPP^+^-mediated cell death that was blocked by necrostatin-1, neither erastin- or RSL3-mediated ferroptosis, nor the associated lipid peroxidation, were inhibited upon incubation of HT1080 cells and mouse embryonic fibroblasts with this necroptosis inhibitor. Thus, although MPP^+^-mediated cell death shares some features of ferroptosis, including cell death inhibition by ferrostatin-1, DFO, and Trolox, and lipid peroxidation, they seem to be distinct. In support of this hypothesis, N-acetylcysteine (NAC) effectively blocked both erastin- or RSL3-mediated lipid peroxidation and cell death of HT1080 cells and mouse embryonic fibroblasts, but was unable to inhibit lipid peroxidation and cell death induced by MPP^+^ in differentiated SH-SY5Y cells [100]. Therefore, understanding the involvement of ferroptosis in MPP^+^-induced cell death could help in identifying the cellular mechanisms triggered by the MPP^+^ toxin, which is commonly used in PD research.

In another study, Geng et al. measured the cell viability of SH-SY5Y cells exposed to the ferroptosis inducer erastin (20 μM) for 24 h. They found increased cell death following erastin exposure, an effect prevented by the ferroptosis inhibitors Fer-1 (2 μM), DFO (100 μM), and NAC (20 mM) [101]. Sun and colleagues found an upregulation of ACSL4 expression and downregulation of GPX4 in undifferentiated SH-SY5Y cells exposed to 6-hydroxydopamine [6-OHDA (80 μM), a neurotoxin widely used to mimic PD], which are compatible with ferroptosis [102]. These studies support the suitability of SH-SY5Y cells as a model to study the mechanisms underlying ferroptosis in PD. In another study, the relationship between α-syn, ferroptosis, and neurodegeneration in SH-SY5Y cells was analyzed. Cells were exposed to Fer-1, Lip-1, and vitamin E and then co-incubated with ferric ammonium citrate (100–400 μM), after which cell death, lipid peroxidation, ROS generation, GSH and α-syn levels were evaluated. Increased ROS generation and increased malondialdehyde (MDA, a biomarker of lipid peroxidation) levels, together with a reduction in the GSH content, were detected in SH-SY5Y cells exposed to iron. The co-incubation with ferroptosis inhibitors (Fer-1, Lip-1, and vitamin E) prevented the toxic actions of iron (namely the generation of ROS through the Fenton reaction) and prevented the accumulation of α-syn [103].

Similarly, SH-SY5Y cells have been used to explore the involvement of ferroptosis in AD. Cong and colleagues designed and synthesized hydroxylated chalcones with the capacity to inhibit the aggregation of amyloid-β peptides, as well as to simultaneously inhibit ferroptosis. In undifferentiated SH-SY5Y cells, compounds **14**a–c (10 μM, for 48 h), with three hydroxyl substituents, significantly prevented Aβ1-42 aggregation and reversed the inhibition of GPX4 (induced by RSL3) and system Xc- (promoted by erastin), thus inhibiting ferroptosis [104].

SH-SY5Y cells have also been applied to explore the involvement of ferroptosis in HD. SH-SY5Y cells overexpressing pathological huntingtin with polyglutamine expansions showed an increased sensitivity to catecholamine-induced toxicity. Furthermore, the mechanism underlying neuronal degeneration might involve the generation of H_2_O_2_ and ferroptosis, as demonstrated by the attenuation of cell death near to control levels by treating cells with monoamine oxidase (MAO) inhibitors, catalase and with the ferroptosis inhibitor, and deferoxamine (50 μM) [105].

Wang and colleagues [106] used undifferentiated SH-SY5Y cells to explore the role of mitochondrial ferritin (FtMt) on erastin-induced ferroptosis. For that purpose, FtMt overexpressing SH-SY5Y cells (FtMt-SH-SY5Y), control SH-SY5Y cells, and empty vector-SH-SY5Y cells were exposed to erastin (10 μM, for 24 h). A significant decrease in the viability of SH-SY5Y cells and vector-SH-SY5Y cells was observed, along with a decrease (although less significant) in FtMt-SH-SY5Y cells. Given the involvement of the voltage-dependent anion channel (VDAC) in erastin mechanism of action, the expression of VDAC2 and VDAC3 was further analyzed. Increased expression of both VDAC2 and VDAC3 was shown in control SH-SY5Y and vector-SH-SY5Y cells exposed to erastin, but no significant alterations were detected In FtMt-SH-SY5Y cells. This suggests a specific protective effect of FtMt on erastin-induced ferroptosis. In addition, increased levels of the labile iron pool (LIP) and L-ferritin were revealed in control SH-SY5Y and vector-SH-SY5Y cells following erastin treatment, although no significant changes were detected in FtMt-SH-SY5Y cells. This demonstrates a protective effect for FtMt against erastin-induced cell death [106]. Table 1 summarizes the studies that used this cell model to elucidate the involvement of ferroptosis in the pathophysiology of NDs.

#### 4.1.2. PC12 Cell Line

Rat pheochromocytoma PC12 cells have been widely applied in the study of NDs in vitro [107]. Although this cell line originated from a pheochromocytoma of the rat adrenal medulla, PC12 cells have been extensively characterized for neurosecretion (catecholamines, dopamine, and norepinephrine), and for the presence of neurotransmitter receptors [nicotinic acetylcholine receptor and gamma-aminobutyric acid (GABA)] and ion channels (Na^+^ and K^+^) [107].

When cultured in the presence of nerve growth factor (NGF), these cells differentiate morphologically and functionally into sympathetic ganglion neurons [108,109]. Thus, since they synthesize and store dopamine, they represent a suitable in vitro model to study catecholaminergic neurotoxicity [110]. The main advantages and limitations of this cell line are represented in Figure 4 and listed in Appendix A.

A study performed by Wu et al. explored cell death induced by *tert*-butylhydroperoxide (*t*-BHP) in PC12 cells, to mimic the oxidative stress present in PD. They observed an increased production of lipid ROS and decreased GPX4 expression in cells exposed to *t*-BHP (100 μM for 1 h). These effects were significantly reversed by the ferroptosis inhibitors Fer-1 (1 μM, pre-treatment for 24 h) and DFO (100 μM, pre-treatment for 24 h) [111]. Another study explored the involvement of ferroptosis-related mechanisms in the MPP^+^ effects on PC12 cells. Exposure of PC12 cells to MPP^+^ (0.4 mM) for 24 h induced significant cell death, decreased the expression of GPX4, and reduced intracellular GSH levels. Importantly, the MPP^+^ toxic effects were attenuated in cells pre-treated with the ferroptosis inhibitor Fer-1 (2 μM, for 30 min before MPP^+^) [112].

Xi et al. analyzed the potential of Hinokitiol (Hino) to act as a potent ferroptosis inhibitor, namely its ability to rescue from neuronal damage both in vitro, in PC12 cells, and, in vivo, in zebrafish. First, the group exposed PC12 cells to the ferroptosis inducer RSL3 (3 μM, for 12 h) and concluded that deferiprone and deferoxamine did not exhibit any protection against RSL3-induced cell death. However, Hino (20 μM, for 4 h) showed a higher protective activity, rescuing PC12 cells from RSL3-induced cell death. In PC12 cells exposed to 6-OHDA (a model of PD), pre-treatment with Hino (20, 40, and 60 μM for 4 h) significantly attenuated cell death and rescued morphological damages induced by 6-OHDA (400 μM for 3 h). In addition, 6-OHDA (400 μM, for 3 h) treatment increased both ROS levels and lipid peroxidation, which was inhibited upon pre-treatment with Hino (40 μM, for 4 h). When evaluating iron metabolism, it was observed that the expression of Transferrin receptor 1 (TfR1) and ferroportin (FPN) were significantly decreased in 6-OHDA-treated cells, while the expression of ferritin heavy chain (FTH) was increased. On the other hand, Hino was able to mitigate the 6-OHDA-induced alterations in TfR1, FPN, and FTH levels, in a concentration-dependent manner. In relation to ferroptosis features, Hino upregulated mRNA levels of SLC7A11 prevented GSH loss induced by 6-OHDA or RSL3, and upregulated GPX4 expression. This demonstrates that Hino neuroprotective activity might be related to the inhibition of ferroptosis. The results obtained in zebrafish (exposed to 250 μM of 6-OHDA for 2 h, and Hino, at concentrations of 1, 2, and 4 μM for 24 h) were in agreement with those obtained in PC12 cells regarding the expression of ferroptosis-related genes upon exposure to 6-OHDA and pre-treatment with Hino, namely the upregulation of gpx4a, hmox1, slc7a11, tfr1a, fthl27, and slc40a1 [113].

A study performed by Zhang in 2022 analyzed, in PC12 cells, the protective effects of senegenin against cytotoxic injury induced by Aβ25-35. It was observed that Aβ25-35 (20 μM 24 h) significantly decreased cell viability and GPX expression, and significantly increased ROS generation and MDA levels, when compared to control cells. Importantly, Fer-1 (5 μM for 24 h) was able to counteract Aβ25-35-mediated damage of PC12 cells. In addition, the expression of ACSL4 and Phosphatidylethanolamine binding protein 1 (PEBP1) proteins were elevated in the Aβ25-35-treated cells, with a concomitant decrease in GPX4 expression, features that were reversed upon treatment with Fer-1. Co-exposure of PC12 cells to Aβ25-35 and senegenin (60 μM for 24 h) significantly reduced ROS generation and MDA levels, and increased GPX4 levels, when compared with Aβ25-35-treated cells. Furthermore, the expression of ACSL4 and PEBP1 proteins decreased after senegenin treatment. These results indicate that senegenin remarkably attenuates the oxidative damage caused by Aβ25-35 and also affects the expression of ferroptosis-related proteins, thus significantly protecting PC12 cells against Aβ25-35-induced cytotoxicity [114].

In the same in vitro model, Huang and colleagues recently demonstrated the ability of the neurotoxin 6-OHDA (40 µM, for 24 h) to induce ferroptosis, as observed by a significant decrease in GPX4 expression and the GSH/GSSG ratio, and by a significant increase in MDA levels, thus suggesting increased lipid peroxidation. Importantly, 6-OHDA-mediated changes in the expression of GPX4 were further aggravated upon co-treatment of PC12 cells with 6-OHDA and erastin (1 µM, 24 h) (more pronounced decrease in GPX4 expression when compared to cells treated only with 6-OHDA). In addition, the ferroptosis inhibitor Fer-1 (25 µM, 24 h) was capable of counteracting 6-OHDA-mediated ferroptosis, significantly inhibiting 6-OHDA-mediated decrease in GPX4 expression [115].

Tang et al. studied the neuroprotective effect of 1,6-O,O-diacetyl-britannilactone (OABL, 1, 5, and 10 µM) in PC12 cells exposed to H_2_O_2_ (0.6 mM), 6-OHDA (1.0 mM), glutamate (100 mM), and RSL3 (10 µM) for 24 h. Co-treatment of PC12 cells with OABL significantly rescued the cell injury caused by the chemical aggressors [116].

A summary of the studies where this cell model was used to elucidate the involvement of ferroptosis in the pathophysiology of NDs is presented in Table 1.

#### 4.1.3. LUHMES Cell Line

The lund human mesencephalic (LUHMES) immortalized cell line has been frequently applied to study the pathophysiological mechanisms underlying NDs and also to study drug-induced neurotoxicity. Cells were obtained from the embryonic mesencephalic tissue from an 8-week-old human embryo. It was immortalized through the insertion of the v-myc transgene under the control of a tetracycline-responsive promoter [117]. These cells are compatible with large-scale cultures and present a great consistency between cells in different experiments, which allows the development of useful assays for the screening of both neuroprotective compounds and cellular/molecular targets. They can be differentiated with tetracycline, cyclic AMP (cAMP), and glial-derived neurotrophic factor (GDNF) into a dopaminergic neuronal phenotype, expressing specific neuron markers [tyrosine hydroxylase (TH), DAT, synapsin I (SYN1), potassium inwardly-rectifying channel, synaptophysin (SYP), tubulin beta-3 (Tuj3), and nuclear receptor related 1 (Nurr1)] [118,119]. The fact that these cells exhibit spontaneous electrical activities and are able to release/uptake dopamine, indicates the high applicability of differentiated LUHMES cells in the study of dopaminergic neurodegeneration, as these cells may closely resemble human dopaminergic neurons. Therefore, differentiated LUHMES cells are recognized as a robust neuronal model, suitable for in vitro studies regarding disease modelling, neurodevelopment, and neuropharmacology [119]. The main advantages and limitations of this cell line are shown in Figure 4 and listed in Appendix A.

Using LUHMES cells as an in vitro model of Parkinson’s disease, Gutbier and colleagues evaluated the neuroprotective effects of seven synthetic compounds (with a structure similar to iron chelators) against the ferroptosis induced by erastin (1.25 µM for 24 h). Similarly to DFO (200 µM for 24 h), two of the tested compounds (50 µM, 24 h) conferred neuroprotection against erastin-induced cell death. Thus, in this cell model, the authors demonstrated the effectiveness of iron chelators in reversing this type of ferroptotic cell death [120].

Another study developed in 2016 concluded that ferroptosis was a remarkably effective pathway for inducing the death of LUHMES cells. Indeed, the group observed that erastin was an extremely potent inducer of LUHMES cells’ death, at concentrations ranging from 2 to 6 µM, and that the observed effects involved GSH depletion. In addition, the ferroptosis inhibitors DFP and NAC (100 µM and 10 µM, respectively, 1 h after erastin exposure) attenuated the erastin-induced cell death in this cell model [121]. This supports the usefulness of using this cellular model to study the impact of ferroptosis on neurodegeneration.

Another study clarified the susceptibility of LUHMES cells to ferroptosis. The group exposed cells to the ferroptosis inducers erastin (12.5 µM, for 4 h), RSL3 (0.625 µM, for 4 h), and ML210 (0.625 µM, for 4 h), and observed that all compounds significantly reduced cell viability, which was reversed by Fer-1 treatment (5 µM, 4 h). Lipid peroxidation was also shown in LUHMES cells exposed to all the ferroptosis inducers, and Fer-1 significantly prevented this effect. To examine the role of labile metals in neuronal ferroptosis, metal chelators were tested at noncytotoxic chelator concentrations to understand if any compound can protect LUHMES neurons from the cytotoxicity induced by erastin or RSL3. Deferoxiprone (DFX, 10 µM, 24 h), 2,3-dimercapto-1-propanesulfonic acid (DMPS, 50 µM, 24 h), and DFP (50 µM, 24 h) significantly protected LUHMES neurons from the cytotoxicity induced by both erastin and RSL3 [122]. A summary of the studies where this cell model was used to elucidate the involvement of ferroptosis in several NDs is presented in Table 1.

#### 4.1.4. HEK293 Cell Line

Another common in vitro model used in NDs research is the HEK293 cell line (human embryonic kidney 293). This cell line was established from the kidney of an aborted human embryo, which after transfection with sheared adenovirus 5 DNA, stably expresses the adenoviral E1A and E1B-55k proteins. The incorporation of the adenovirus genome into chromosome 19 is known to interfere with the cell cycle pathways and apoptosis [123]. The main advantages and limitations of this cell line are listed in Figure 4 and Appendix A.

A study carried out by Shah et al. used HEK293 cells to up-regulate LOX expression and to assess cell sensitivity to ferroptosis in PD. They found a positive correlation between LOX overexpression and RSL3-induced ferroptosis, decreasing the Median lethal dose (LD50) from 6.8 µM in the wild-type cells to 0.6 µM in the 5-LOX overexpressing cells [124]. LOX has a key role in lipid peroxidation by promoting the peroxidation of AA/AdA-PE to AA/AdA-PE-OOH, leading to ferroptosis [47].

The HEK293 cell line has also been used to study ferroptosis involvement in Huntington’s disease. A study performed in 2018 analyzed the expression of iron regulatory protein (IRP1) in HEK293 cells expressing the N-terminal of the huntingtin-coding gene, containing 20 or 160 polyglutamine repeats. IRP1 post-transcriptionally regulated the homeostasis of intracellular iron levels, binding to iron responsive elements implicated in iron metabolism. An increase in IRP1 at both the protein and messenger ribonucleic acid (mRNA) levels was observed in cells expressing the huntingtin version with 160 polyglutamine repeats, as compared to the huntingtin version with 20 polyglutamine repeats. Furthermore, the expression of transferrin (Tf), TfR, and ferritin (proteins involved in iron homeostasis) were also significantly increased. Therefore, these results indicate that mHtt may possibly disturb iron homeostasis (a feature of ferroptosis) by disturbing the expression of iron regulatory proteins [125]. A summary of the studies where this cell model has been used to evaluate the involvement of ferroptosis in several NDs is presented in Table 1.

#### 4.1.5. NSC-34 Cell Line

The NSC-34 cell line is a murine neuroblastoma/spinal cord hybrid cell line obtained through the fusion of mouse neuroblastoma cells with motoneuron-enriched embryonic spinal cord cells [126]. The NSC-34 cell line has been used for many years in studies of motor diseases. However, the lack of glutamate toxicity represents a limitation of this cell line, for example in the study of ALS (glutamate-mediated excitotoxicity is characteristic of ALS and the absence of glutamate toxicity reported for this cellular model limits its use in the study of this pathology). To bypass these limitations, NSC-34 cells can be differentiated into a glutamate-sensitive neuronal cell population. The differentiation allows cells to express a high degree of properties of motor neurons, which includes the formation of contacts with cultured myotubes, release of neurotransmitters, expression of choline acetyltransferase, and synthesis and storage of acetylcholine [127]. The main advantages and limitations of the NSC-34 cell line are represented in Figure 4 and listed in Appendix A.

A study performed by Martinez and colleagues addressed the sensitivity of NSC-34 cells to ferroptosis. Cells were treated with erastin and RSL3 together with ferric citrate (50 μg/mL). This combination induced the death of NSC-34 cells, which was suppressed by Fer-1 treatment. In addition, an increase in lipid peroxidation upon erastin + RSL3 treatment was observed, a feature of ferroptosis [128]. A summary of the studies where this cell model was used is presented in Table 1.

#### 4.1.6. Organoids

Another in vitro model that has been successfully used to investigate ferroptosis in neurodegeneration is the organoid. An organoid can be defined as “a collection of organ-specific cell types that develops from stem cells or organ progenitors and self-organizes through cell sorting and spatially restricted lineage commitment in a manner similar to in vivo” [129]. The three-dimensional (3D) brain organoids are self-organized and spontaneously differentiated from human pluripotent stem cells, and are composed of progenitor, neuronal, and glial cell types, analogous to the fetal human brain. Accordingly, brain organoids are different from conventional two-dimensional cell cultures since they replicate the structure of the human brain at the cellular level, as well as the developmental trajectory of the brain. Thus, they enable the simulation of the development and function of the brain, which are often inaccessible to direct experimentation [130]. On the other hand, the low reproducibility, the requirement for improved differentiation protocols, and ethical concerns represent important drawbacks of this model [92]. The main advantages and limitations of organoids are represented in Figure 4 and listed in Appendix A.

A previous study explored the mechanisms of ferroptosis in neuronal organoids [three-dimensional neuronal organoids generated with differentiated LUHMES cells, and using erastin (10 μM, 24 h) as a ferroptosis inducer]. Protection against erastin-induced ferroptosis by Fer-1 (100 nM), desferoxamine (5 μM) and SK4 (100 μM, a compound with a similar structure to iron chelators) was observed after 24 h [120]. This demonstrates the suitability of brain organoids for modelling ferroptosis in vitro.

A recent experimental approach was implemented by coupling cerebral organoids to a perfusion system that closely mimics the cortical vasculature. The obtained vascularized human cortical organoids (vhCOs) demonstrated the advantage of overcoming the limitations of conventional organoids (as mentioned above). This in vitro model contains the classical blood-brain barrier (BBB) properties, including the expression of tight junction proteins and the presence of a high transendothelial electrical resistance, mimicking the brain more clearly [131,132].

Although organoids are still not frequently used for the study of ferroptosis, they have gained increasing relevance in the study of other pathophysiological mechanisms of NDs, which allows us to predict that their use may be of great importance to study the ferroptosis impact in neurodegeneration. A summary of the studies where organoids have been used to explore ferroptosis in NDs is presented in Table 1.

#### 4.1.7. Induced Pluripotent Stem Cell Line

Induced pluripotent stem cell lines (iPSCs), obtained from patients with incurable diseases, are a promising approach for studying the mechanisms underlying several diseases, and also for drug development and screening [133]. The main advantages and limitations of in vitro models based on induced pluripotent stem cells are represented in Figure 4 and listed in Appendix A.

Some studies have been exploring ferroptosis in iPSCs. Angelova et al. studied the influence of α-syn aggregation on ferroptosis in iPSC cells. They found that increased superoxide production and lipid peroxidation (features of ferroptosis) were involved in the cell death detected in iPSC-derived neurons with the α-syn triplication (SNCA x3, which results in four copies of the SNCA gene, and a doubling of SNCA mRNA and α-syn protein), suggesting an activation of ferroptosis. In accordance with this hypothesis, ferroptosis inhibition with DFO, D4-Lnn (inhibitor of lipid peroxidation), and Fer-1 reduced oligomer-induced toxicity in this cell model [134]. This study shows the usefulness of iPSC as a model to study ferroptosis in PD.

Kondo et al. generated iPSCs from familial and sporadic AD patients and differentiated them into neural cells. They found that Aβ oligomers were significantly accumulated in iPSC-derived neurons with the amyloid precursor protein (APP)-E693∆ mutation (rare, autosomal-recessive mutations of the gene coding for APP and related to familial AD), leading to oxidative stress [135]. Considering the role of oxidative stress in ferroptosis, this suggests that iPSCs have the potential to be used for exploring ferroptosis in AD research.

Matsuo and colleagues focused on ferroptosis in human hiPSC-derived motor neurons to study ALS. Accordingly, in hiPSC-derived motor neurons exposed to the GPX4 inducer-RSL3 for 24 h, a concentration-dependent cell death was observed (concentrations ranging from 0.01 to 10 µM). In addition, the RSL3-induced cell death of hiPSC-derived motor neurons was prevented by the antioxidant vitamin E acetate, by the lipid peroxidation inhibitor Fer-1 (1 µM, 24 h), and by the iron chelator DFO (100 µM, 24 h), pointing to a role for ferroptosis in ALS [136]. A summary of the studies where this cell model was used in the study of ferroptosis impact in NDs is presented in Table 1.

#### 4.1.8. Primary Neuronal Cultures

Primary neuronal cultures are an in vitro model extensively used to dissect the molecular and cellular mechanisms that underlie human brain diseases.

Primary dopaminergic neurons represent an extremely valuable in vitro model in NDs research. These primary neurons are generally obtained from embryonic murine brain tissue, which rapidly differentiate in culture and form neurites and synapses [137,138]. The main advantages and limitations of this model are represented in Figure 4 and listed in Appendix A.

Using primary dopaminergic neurons, Zhang et al. studied the deleterious effects of the ferroptosis inducer erastin. They found a decreased viability, increased generation of ROS, and downregulation of GPX4 and system Xc- expression in primary dopaminergic neurons exposed to erastin (50 µM) for 48 h. All these effects were reversed by treating cells with deferoxamine (50 µM, 12 h before erastin treatment, plus 48 h) [57]. This suggests that the mechanisms involved in ferroptosis are conserved in primary dopaminergic neurons, representing, therefore, a good system to study ferroptosis in neurodegeneration. A summary of the studies where this cell model was used in the study of ferroptosis involvement in NDs is presented in Table 1.

#### 4.1.9. Glial Cell Culture

Glial cells, including astrocytes, oligodendrocytes, and microglia, play a crucial role in the normal functioning and development of the human brain, providing structural, trophic, and metabolic support to neurons. In addition, they are also involved in the uptake and synthesis of neurotransmitters, immunomodulation, adult neurogenesis, regulation of synaptogenesis, and synaptic plasticity. Accordingly, potential misfunctions of glial cells can ultimately lead to neurological disorders such as NDs [139]. Therefore, the use of distinct in vitro models of glial cells can elucidate their involvement in the onset and progression of NDs and may represent potential new targets for disease-modifying drugs for the treatment of such conditions.

Astrocytes are the most common type of glial cells, which present a variety of functions remarkably important for CNS homeostasis, such as the modulation of synaptic activity, maintenance of the BBB integrity, regulation of the extracellular iron balance, activation of the neuroinflammatory response and support of neuronal activity [140]. Primary astrocyte cultures are one of the most common models in NDs research, as they are suitable for repeated passages and amenable to cryopreservation. On the other hand, their growth is relatively slow, which makes them difficult to obtain [141]. Astrocyte cell lines include CTX TNA2, 9L/lacZ, and N1E-115 cells, among others.

Some studies have been using primary astrocytes to study ferroptosis. A study performed in 2021 used mouse primary astrocytes from the cortex stimulated with angiotensin II (10 mM), in the presence or absence of Fer-1 (1 or 2 µM). The inflammatory markers interleukin-1 beta (IL-1β), cyclooxygenase-2 (COX-2), and interleukin-6 (IL-6) were significantly elevated in the astrocytes upon stimulation with angiotensin II. In addition, angiotensin II caused an overproduction of ROS, a reduction in GSH levels, and the downregulation of GPX4, Nrf2, and heme oxygenase (HO-1). All these effects of angiotensin II in astrocytes were reversed by Fer-1 (concentration-dependent effect) [142].

Microglia are brain cells responsible for the maintenance of neuronal networks, for the regulation of brain development and injury repair, and are recognized as the macrophages of the brain. They are also responsible for the elimination of dead cells, reductant synapses, protein aggregates, microbes, and antigens that may compromise the CNS. Furthermore, as the principal source of pro-inflammatory cytokines, microglia are essential in neuroinflammation and also are able to induce cellular responses [143]. The activation of microglia is observed in active demyelinating MS lesions, pre-active lesions, and areas of remyelination [144]. Numerous isolation and culture protocols for microglia have been developed, although the use of rapid isolation methods is crucial to uncover the initial trigger of microglia activation [141]. To circumvent the limitations in obtaining primary cultures of microglia, immortalized human microglia cell lines have been established. Microglia cell lines include HAPI, BV-2, EOC2, C8-B4, CHME-5, and HMO6-. The main advantages and limitations of microglia cell cultures are represented in Figure 4 and listed in Appendix A.

**Table 1 pharmaceutics-15-01369-t001:** Compilation of different studies using distinct in vitro models for the evaluation of ferroptosis’ involvement in neurodegenerative diseases.

Model	Compounds	Main Conclusions	Reference
**SH-SY5Y cells**	Aggressor: **MPP**^+^ (5 mM, 48 h) Ferroptosis inhibitors: **Fer-1** (1–5 μM, 48 h), **DFO** (100–500 μM, 48 h) and **Trolox** (100–500 μM, 48 h)	MPP^+^-induced cell death was blocked by ferroptosis inhibitors.	[100]
	Aggressor: **Erastin** (20 μM, 24 h) Ferroptosis inhibitors: **Fer-1** (2 μM, 24 h), **DFO** (100 μM, 24 h) and **NAC** (20 mM, 24 h)	Increased cell death promoted by erastin, which was reversed by ferroptosis inhibitors.	[101]
	Aggressor: **ferric ammonium citrate** (100–400 μM) Ferroptosis inhibitors: **Fer-1**, **Lip-1** and **vitamin E**	The aggressor induced an increase in the generation of ROS and in the MDA levels, with a concomitant decrease in the GSH content. The ferroptosis inhibitors counteracted all these effects.	[103]
	Aggressor: **Erastin** (10 μM, 24 h) Ferroptosis inhibitor: **Mitochondrial Ferritin**	FtMt protected cells against erastin-induced cytotoxicity.	[106]
**PC12 cells**	Aggressor: ***t*-BHP** (100 μM, 1 h) Ferroptosis inhibitors: **Fer-1** (1 μM, 24 h) and **DFO** (100 μM, 24 h)	*t*-BHP promoted ROS generation and GPX4 decreased expression. These parameters were reversed by ferroptosis inhibitors.	[111]
	Aggressor: **MPP^+^** (0.4 mM, 24 h) Ferroptosis inhibitor: **Fer-1** (2 μM, 30 min before the aggressor)	The aggressor decreased GPX4 expression and reduced GSH levels. The ferroptosis inhibitor attenuated MPP^+^-induced cytotoxic effects.	[112]
	Aggressor: **RSL3** (3 μM, 12 h) and **6-OHDA** (400 μM, 3 h) Ferroptosis inhibitor: **Hino** (20, 40 and 60 μM, 4 h)	Hino protected cells against RSL3-induced cell death. Hino protected cells against 6-OHDA-induced cell death and rescued cells from mitochondrial damage. Also, it reversed the increased ROS production and lipid peroxidation promoted by 6-OHDA, and increased GSH, GPX4 and SLC7A11 expression levels. Hino augmented the expression of TfR1 and FPN, and decreased FTH.	[113]
	Aggressor: **Aβ25-35** (20 μM, 24 h) Ferroptosis inhibitor: **senegenin** (60 μM, 24 h) and **Fer-1** (5 μM, 24 h)	Senegenin attenuated the oxidative damage caused by Aβ25-35 and also affected the expression of ferroptosis-related proteins.	[114]
	Aggressor: **6-OHDA** (40 µM, 24 h) Ferroptosis inhibitor: **Fer-1** (25 µM, 24 h)	6-OHDA induced ferroptosis, and erastin aggravated the effects. Fer-1 was capable of counteracting 6-OHDA-mediated ferroptosis.	[115]
	Aggressor: **H_2_O_2_** (0.6 mM, 24 h), **6-OHDA** (1.0 mM, 24 h), **Glutamate** (100 mM, 24 h) and **RSL3** (10 µM, 24 h) Ferroptosis inhibitor: **OABL** (1, 5 and 10 µM, 24 h)	Co-treatment of PC12 cells with OABL significantly rescued the cell injury caused by the chemical aggressors.	[116]
**LUHMES cells**	Aggressor: **Erastin** (1.25 μM, 24 h) Ferroptosis inhibitors: **DFO** (200 μM, 24 h)	Erastin induced a significant cell death, which was reverted by DFO.	[120]
	Aggressor: **Erastin** (2–6 μM) Ferroptosis inhibitors: **DFP** (100 μM, 1 h) and **NAC** (10 mM, 1 h)	Erastin was an extremely potent induced of cell death. Ferroptosis inhibitors attenuated the erastin-induced cell death.	[121]
	Aggressor: **Erastin** (12.5 mM, 4 h), **RSL3** (0.625 mM, 4 h) and **ML210** (0.625 mM, 4 h) Ferroptosis inhibitors: **Fer-1** (5 μM, 4 h), **deferoxiprone** (10 mM, 24 h), **DMPS** (50 mM, 24 h) and **DFP** (50 mM, 24 h)	The ferroptosis inducers reduced cell viability and promoted lipid peroxidation, but Fer-1 reverted these effects. DFX, DMPS and DFP protected LUHMES neurons against RSL3-induced cell death.	[122]
**HEK293 cells**	Aggressor: **RSL3**	It was found a positive correlation between LOX overexpression and RSL3-induced ferroptosis, decreasing the LD_50_ from 6.8 µM in the wild-type cells to 0.6 µM in the 5-LOX overexpressing cells.	[124]
**NSC-34 cells**	Aggressor: **RSL3 + ferric citrate** (50 μg/mL) Ferroptosis inhibitor: **Fer-1**	This combination induced a significant cell death, which was suppressed by Fer-1.	[128]
**Organoids**	Aggressor: **Erastin** (10 μM, 24 h) Ferroptosis inhibitors: **Fer-1** (100 nM, 24 h), **DFO** (5 μM, 24 h) and **SK4** (100 μM, 24 h)	Erastin induced a significant cell death, which was reversed by ferroptosis inhibitors.	[120]
**iPSCs**	Aggressor: **RSL3** (0.01 to 10 μM, 24 h) Ferroptosis inhibitors: **Vitamin E**, **Fer-1** (1 μM, 24 h) and **DFO** (100 μM, 24 h)	A concentration-dependent cell death was detected after exposure to the aggressor. Ferroptosis inhibitors reversed the aggressor-mediated cell death.	[136]
**Primary neuronal cultures**	Aggressor: **Erastin** (50 μM, 48 h) Ferroptosis inhibitor: **DFO** (50 μM)	Erastin promoted a decreased cell viability, increased the generation of ROS and downregulated GPX4 and system Xc- expression. All these effects were reversed by DFO.	[57]
**Glial cell culture**	Aggressor: **Angiotensin II** (10 μM) Ferroptosis inhibitors: **Fer-1** (1 or 2 μM)	Angiotensin II caused an overproduction of ROS, a reduction in GSH levels, and the downregulation of GPX4, Nrf2 and heme oxygenase, all reverted by Fer-1.	[142]
	Aggressor: **GODs** (100 μg/mL, 24 h) Ferroptosis inhibitors: **Fer-1** and **DFO**	GODs promoted ferroptosis, which was reversed by Fer-1 and DFO.	[145]
	Aggressor: **hemin** (10–100 μM, 12 and 24 h) Ferroptosis inhibitors: **Fer-1** (2 μM, 12 and 24 h) and **DFO** (100 μM, 12 and 24 h)	Hemin promoted an increase in ROS and MDA levels and induced the upregulation of genes involved in ferroptosis (such as the *SLC7A11* gene). Ferroptosis inhibitors rescued cells from cell death.	[146]

6-OHDA: 6-hydroxydopamine; DFO: Deferoxamine; DFP: Deferiprone; DFX: deferoxiprone; DMPS: 2,3-dimercapto-1-propanesulfonic acid; Fer-1: Ferrostatin-1; FPN: Ferroportin; FTH: Ferritin heavy chain; FtMt: Mitochondrial Ferritin; GPX4: Glutathione peroxidase 4; GSH: Glutathione; GODs: graphene quantum dots; iPSCs: Induced pluripotent stem cell lines; Lip-1: Liproxstatin-1; MDA: Malondialdehyde; MPP^+^: 1-Methyl-4-phenylpyridinium; NAC: N-acetylcysteine; OPCs: Oligodendrocyte precursor cells; ROS: Reactive oxygen species; *t*-BHP: *Tert*-butylhydroperoxide; TfR1: Transferrin receptor 1.

Similar to astrocytes, microglia have been used to study ferroptosis. In a study performed in 2020, ferroptosis activation by graphene quantum dots (GODs) was evaluated in BV2 cells. The group concluded that GODs (100 µg/mL, 24 h) induced ferroptosis in this cell model, which was reversed by pre-treatment with Fer-1 and DFO (ferroptosis inhibitors). In addition, the pre-treatment of the cells with the two ferroptosis inhibitors reduced the cytosolic iron content [145]. Despite the strong correlation between microglia and neurodegeneration, the involvement of ferroptosis in this cell type is not yet evident. However, the available data indicate microglia as a suitable model to study ferroptosis.

Oligodendrocytes are the cells responsible for myelin production in the CNS. Developing myelin and demyelination following damage depends on the proper function of oligodendrocytes, such as the ability of oligodendrocyte precursor cells (OPCs) to migrate to the damaged area, to differentiate into mature myelin-forming cells, and to maintain axon health. Primary rodent oligodendrocytes are one of the most used models, which can be maintained in vitro for several weeks [141]. Furthermore, culture systems based on human oligodendrocytes or OPCs have also been widely used [141]. These models can be differentiated and are able to produce myelin, in vitro, in the absence of signals from axons. Rat OPCs are easier to isolate and maintain. However, human OPCs take longer to develop the oligodendrocyte phenotype and need to be isolated from brain biopsies (which are associated with ethical problems) [92]. Oligodendrocyte cell lines include the CG4, OLN-93, Oli-Neu, N19/N20.1, HOG, and MO3-13 cell lines.

Some studies have been using oligodendrocytes to study ferroptosis. A study performed by Shen and colleagues treated cells with hemin (an in vitro model of the hemorrhagic model). They showed increased levels of ROS and MDA (a lipid peroxidation biomarker) in hemin-treated cells, suggesting ferroptosis activation (in concentrations between 10 and 100 µM, for 12 and 24 h). In addition, hemin induced a time-dependent upregulation of genes involved in ferroptosis, including the *SLC7A11* gene, which codes for system Xc-. To confirm the involvement of ferroptosis, cell death inhibitors were tested: Fer-1 (2 µM) and DFO (100 µM) as ferroptosis inhibitors, Necrostatin-1 (Nec-1, 100 µM) as a necroptosis inhibitor, 3-methyladenine (3-MA, 1 mM) as an autophagy inhibitor, and QVD (10 µM) as an apoptosis inhibitor, for 12 and 24 h. Only the ferroptosis inhibitors (Fer-1 and DFO) rescued OPCs from cell death. These ferroptosis inhibitors also decreased the levels of lipid ROS induced by hemin treatment [146].

Although no studies have been using oligodendrocytes in vitro to study ferroptosis in NDs, the available data suggest a potential for this cell model at this level. A summary of the studies where this cell model was used in the study of ferroptosis involvement in NDs is presented in Table 1.

#### 4.1.10. Brain Slices

Several 3D organotypic brain slices obtained from rodents have been extensively applied to study the pathogenesis of different NDs, since they can be obtained from different areas of the CNS. This allows experimental manipulations that are complicated to perform in vivo. In the case of MS, the brain slices can be demyelinated, in vitro, upon exposure to toxins, aiming at the discovery of new drugs capable of counteracting neuronal demyelination. However, numerous compounds that induced remyelination and increased neuronal survival in rodent models of MS have failed when tested in human clinical trials [92,147]. However, brain slices can be used to study other NDs, being a good model that approximates the in vitro to the in vivo.

Although there are no studies evaluating ferroptosis in brain slices, this in vitro model has all the characteristics to make this connection possible. The main advantages and limitations of brain slices are listed in Figure 4 and Appendix A.

### 4.2. In Vivo Models

Ferroptosis is an emerging field of research regarding its potential involvement in the pathophysiology of distinct NDs. However, given its relatively recent discovery and correlation with NDs pathophysiology, only a few in vivo studies have been conducted in this research field. Accordingly, the development of appropriate, robust, and reliable in vivo models is mandatory, as it will prompt towards a better understanding of the mechanism underlying neurodegeneration, as well as it allows the identification of potential targets, ultimately leading to the discovery of new disease-modifying drugs capable of stopping or delaying disease progression.

The main advantages and limitations of the in vivo models that are currently the most used in the study of NDs are represented in Figure 5 and listed in Appendix A. A summary of the studies where these models have been used to study the ferroptosis impact in NDs is presented in Table 2.

#### 4.2.1. Rodents

Rodents represent gold standard models in the validation of disease mechanisms and for providing important preclinical data on therapeutic drug targets. Therefore, rodent models have been increasingly used to study NDs.

Transgenic animals have been used in different protocols, as it is believed that they express some common characteristics of both genetic and sporadic forms of several NDs [148].

Mice genetically engineered to develop a loss of dopaminergic neurons in the SN are the most used in vivo model for PD. Such models possess modifications in α-syn, leucine rich repeat kinase 2 (LRRK2), Parkin (Park2), PTEN-induced putative kinase 1 (PINK1), or DJ-1 (Parki7) [149]. Additionally, 6-OHDA, MPTP, lipopolysaccharide, paraquat, rotenone, and manganese are toxins that are widely utilized in rats, mice, cats, and monkeys to mimic the PD phenotype and are, therefore, useful for the study of this neurodegenerative disease [150,151]. A study performed by Ayton et al. showed an elevation of iron in the CNS of an APP knockout mice model (APP-/-). To confirm if the neuronal degeneration observed in APP-/- mice was induced by an iron-dependent mechanism, 3-month-old APP-/- mice were treated with the iron chelator DFP (50 mg/kg/day in drinking water for 3 months). Ameliorated SN neuron loss was detected upon DFP treatment (preventing 50% of the neuronal loss), thus supporting, in vivo, a role for ferroptosis in PD pathogenesis [152].

A study developed in 2022 used Nrf2-/- and Nrf2+/+ mice, at an age of 3 and 6 months old, overexpressing human α-syn (hα-Syn+:Nrf2-/- and hα-Syn+:Nrf2+/+), to evaluate whether the loss of Nrf2 increased distinct biomarkers of ferroptosis in brain regions relevant for PD. The group observed a significant increase of ferrous iron in the midbrain and striatum of hα-Syn+:Nrf2-/- mice (especially at 6 months of age), when compared to the hα-Syn+:Nrf2+/+ mice. In the midbrain, ferrous iron colocalized with tyrosine hydroxylase (TH)-positive neurons, suggesting ferroptosis activation in dopaminergic neurons. Since LC3-II, Nuclear receptor coactivator 4 (NCOA4), and FTH1 levels were elevated in both brain regions of 6-month-old hα-Syn+:Nrf2-/- mice, when compared with hα-Syn+:Nrf2+/+ mice, it suggests that the increased levels of free iron may be the consequence of ferritinophagy blockage. Additionally, 6-month-old hα-Syn+:Nrf2-/- mice showed increased levels of ROS, MDA, and free iron, and reduced levels of manganese superoxide dismutase (SOD2/MnSOD) and GSH, as compared to hα-Syn+:Nrf2+/+ animals. In addition, 6-month-old animals showed significantly increased levels of both oligomeric and phosphorylated α-syn, as compared to 3-month-old animals. This suggests that the observed age-dependent increase in ferroptosis markers was directly correlated with the severity of α-syn pathology [153].

For AD, many rodent models are used, being genetically altered to promote the pathological mechanisms of the disease. The vast majority of experimental models are transgenic mice expressing human proteins involved in the formation of amyloid plaques (β-amyloid peptide) and neurofibrillary tangles (Tau protein) [154]. The most common transgenic mouse models of AD include the PDAPP mouse (expresses high levels of human APP), the Tg2576 mouse [overexpresses a mutant form of APP (isoform 695) with the Swedish mutation (KM670/671NL)], the 3xTg mouse (expresses three mutations: APP KM670/671NL Swedish mutations, Tau P301L, and PSEN1 M146V), the 5xFAD mouse [expresses human APP and PSEN1 transgenes with a total of five AD-linked mutations: the Swedish (K670N/M671L), Florida (I716V), and London (V717I) mutations in APP, and the M146L and L286V mutations in PSEN1), the PS19 mouse (expresses the P301S mutant version of Tau), and the E3FAD and E4FAD models (crosses between the 5xFAD mice, and the APOE3 and APOE4 targeted replacement mice), some of which are already used for the study of ferroptosis (as shown below) [155]. A study performed in 2014 explored if the administration of the ferroptosis inhibitor zileuton (200 mg/L of drinking water for three months) to age triple-transgenic mice [3xTg mice, harboring the APP Swedish mutation (KM670/671NL), the M146V human mutant PS1, and the Tau P301L] could reduce Aβ deposition. They detected significantly lower levels of Aβ1-42 in the brain homogenates of zileuton-treated animals, when compared to the control group. In addition, they observed a significant reduction in the amount of Aβ peptides deposited in the brain of zileuton-treated animals (2000 amyloid burden when compared to almost 6000 in the control group) [156]. These results thus highlighted, in vivo, a potential connection between AD and ferroptosis.

A study performed in 2022 analyzed the anti-AD activity of eriodictyol (50 mg/kg, i.p. for 3 months) using APPswe/PS1E9 transgenic mice. Eriodictyol significantly decreased Aβ aggregation in the brain of APP/PS1 mice and induced a marked reduction of Aβ peptide and p-Tau levels in the brains of eriodictyol-treated APP/PS1 mice. In addition, treatment with eriodictyol caused a marked reduction in the levels of total iron, ferrous iron, TfR, and FTH in the cortex and hippocampus of APP/PS1 mice. On the other hand, while the levels of MDA content were higher in the cortex and hippocampus of APP/PS1 mice, when compared to control mice, treatment with eriodictyol reversed this phenomenon. In addition, increased GPX4 levels were also detected in the hippocampus and cortex of APP/PS1 mice following eriodictyol treatment, when compared to non-treated APP/PS1 mice, demonstrating the protective effect of this compound against ferroptosis [157].

Another study investigated the protective effects of salidroside (50 mg/kg/day i.p.) on alleviating neuronal ferroptosis caused by Aβ1-42 [both wild-type and Nrf2-/- mice were subjected to an intracerebroventricular injection of 2 µL of Aβ1-42 (222 µM) into the right hemisphere; in the Salidroside + Aβ1-42 groups, wild-type mice and Nrf2-/- mice were then treated with salidroside (50 mg/kg/day, for 75 days); a saline solution was administered to control animals] [158]. The mitochondria of Aβ1-42-treated mice were smaller than in the control group, but the treatment with salidroside attenuated these morphological changes. In addition, the treatment with salidroside significantly prevented the decrease in GPX4 expression caused by Aβ1-42 in the hippocampus and cortex of Aβ1-42-treated wild-type mice, thus suggesting the capability of salidroside to counteract ferroptosis in this model. Furthermore, the cognitive function of wild-type animals administered with Aβ1-42 was significantly enhanced upon treatment with salidroside, and the levels of prostaglandin-endoperoxide synthase 2 were significantly decreased. However, such protective effects for salidroside were not observed in Aβ1-42-treated Nrf2-/- mice. These results suggest that salidroside-mediated neuroprotection may be correlated to its ability to rescue neurons from ferroptosis and that such effects depend on Nrf2. To further explore the involvement of Nrf2 activation in the neuroprotective effects of salidroside, the expression levels of Nrf2-regulated antioxidant proteins, namely NAD(P)H: quinone oxidoreductase 1 (NQO1) and HO-1, were also evaluated. Accordingly, salidroside caused a significant upregulation of both HO-1 and NQO1 in wild-type animals treated with Aβ1-42, but not in Nrf2-/- mice. Overall, these results demonstrated that salidroside-mediated neuroprotection against Aβ1-42 depends on the Nrf2 signaling pathway [158].

The elucidation of the gene responsible for inducing HD enabled the use of genetic engineering to develop in vivo models of the disease. Reduced levels of GSH and glutathione-S-transferase (GST) were observed in the striatum, cortex, and hippocampus of mice models of HD [83,159]. This suggests that ferroptosis may be involved in HD pathogenesis and that it can be modulated in HD mice models.

The accumulation of iron is one of the key alterations in ALS, highlighting a potential connection to ferroptosis. A study demonstrated an accumulation of iron in the spinal cord of the SOD1G37R transgenic mice model of ALS, suggesting a role for ferroptosis in disease pathogenesis [79]. In another study, SOD1G93A/GPX4 double transgenic mice showed extended lifespan compared with SOD1G93A mice (an ALS model), as well as delayed disease onset and increased motor function (which was attributed to ameliorated spinal motor neuron degeneration and reduced lipid peroxidation) [160].

One of the most relevant in vivo models of MS is the experimental autoimmune encephalomyelitis (EAE) model, as it mimics both the pathological and clinical characteristics of the disease [161]. EAE is characterized by the presence of inflammation, myelin damage, and neurodegeneration. Thus, it became remarkably important for the development and implementation of therapeutic strategies for MS that target the immune system [141]. Studies in the EAE model have explored the activation of ferroptosis mechanisms. In fact, a reduction of mRNA expression of GPX4, total GSH, and system Xc- activity have been detected in this model. These effects are accompanied by increased levels of lipid peroxidation indicators, 4-HNE and MDA [85,162]. Similarly, another study showed reduced GSH and Nrf2 levels in the same model [163].

The cuprizone model represents another type of toxic demyelination, where young adult mice are fed with cuprizone (a copper chelator), leading to a consistent demyelination [164]. Increased levels of 4-HNE, nuclear receptor coactivator 4 (NCOA4, a mediator of ferritinophagy-autophagic degradation of ferritin), and TfR1, and increased COX-2 activity were detected 2 to 4 days after treatment with cuprizone [0.2% (*w/w*)]. On the other hand, GPX4 levels were significantly reduced like the expression of system Xc- [86], suggesting ferroptosis activation in this model. Overall, in vivo models have been supporting the potential involvement of ferroptosis in MS pathogenesis.

Xiang and colleagues evaluated, in a rat model of Temporal lobe epilepsy (TLE) induced by lithium-chloride and pilocarpine (LiCl-Pilo), the potential protective effects of klotho, an anti-aging gene that, when overexpressed, reduces hippocampal neuronal loss and presents neuroprotective activities and anti-cognitive impairment effects in CNS disorders. The overexpression of Klotho was achieved using an adeno-associated viral (AAV) vector delivery system. Through immunofluorescence staining, klotho was shown to mainly localize in neurons rather than astrocytes. In addition, iron accumulation was significantly increased in the hippocampus of AAV-KL (vector harboring klotho) animals, when compared with AAV-NC control rats. Moreover, in the AAV-KL group, the levels of FPN were remarkably higher than in the AAV-NC group, and the opposite occurred for the levels of divalent metal transporter 1 (DMT-1). Finally, the AAV-KL group showed significantly elevated levels of GSH and GPX4 in the hippocampus (compared with the AAV-NC group), but decreased ROS levels. This study demonstrates the involvement of ferroptosis in the neuroprotective mechanism of klotho [165].

Tang et al. studied the neuroprotective effect of 1,6-O,O-diacetyl-britannilactone (OABL) in transgenic mice expressing five familial AD mutations (5xFAD mice). The shrinkage of nuclei and neurons, the accumulation of amyloid plaques, and the high levels of Aβ1-42 and phosphorylated Tau observed in 5xFAD mice were significantly inhibited by OABL (20 mg/kg/day i.p., for 21 days). In addition, OABL increased GSH levels and reduced MDA content in 5xFAD mice, when compared to non-treated animals [116]. Given that a decrease in GSH levels and an increase in MDA content (a lipid peroxidation marker) are two hallmarks of ferroptosis, these results demonstrate the possible protection of OABL against this type of cell death.

Liu et al. analyzed the effects of Mesenchymal stem cell-derived exosomes (MSCs-Exo) on delayed neurocognitive recovery (dNCR) aged mice and evaluated the potential regulatory mechanisms underlying the observed effects. SCs-Exo significantly improved the cognitive impairment of dNCR-aged mice, pointing to a role for exosomes in the treatment of neurological disorders. To determine if the protective effect of MSCs-Exo on cognitive impairment involved ferroptosis inhibition, the group used Fer-1 as a positive control. Compared to the dNCR group, the dNCR+MSCs-Exo and dNCR+Fer-1 groups demonstrated increased mitochondrial size and mitochondrial crest, and reduced density of the double-layer membrane of mitochondria. In addition, administration of MSCs-Exo or Fer-1 to dNCR animals reduced ROS generation, and MDA and Fe^2+^ levels, and increased GSH content, when compared with the dNCR group. These protective effects of MSCs-Exo and Fer-1 involved upregulation of SLC7A11, GPX4, HO-1, and Nrf2 expression, and downregulation of p53. This demonstrates that MSCs-Exo inhibits ferroptosis in the hippocampus of dNCR-aged mice [166].

Huang and colleagues studied the involvement of Sorting Nexin 5 (SNX5, an endosome protein involved in the identification, transport, and unloading of substances between organelles within the cells) in the onset/progression of PD in a rat model of the disease [animals administered with 6-OHDA in the medial forebrain bundle (25 μg)]. This PD model showed a significant decrease in TH and GPX4 levels, and GSH/GSSG ratio. By contrast, MDA levels were significantly increased, suggesting the favoring of lipid peroxidation in this disease model. Furthermore, SNCX5 expression levels were significantly increased in PD rats, suggesting that 6-OHDA-mediated abnormal increases in the SNX5 expression levels and ferroptosis may, potentially, induce the development of PD. Thus, given its ability to induce ferroptosis in PD, SNX5 might represent an alternative target to explore potential therapeutical approaches for PD treatment [115].

Although there are limited studies on the involvement of ferroptosis in NDs, these in vivo models may represent useful tools in the study of such diseases.

#### 4.2.2. Drosophila Melanogaster, Caenorhabditis Elegans, and Zebrafish

The fruit fly *Drosophila melanogaster* and the nematode *Caenorhabditis elegans* have also been used in the investigation of both the cellular and the molecular pathways underlying different NDs. *Drosophila melanogaster* can incorporate modifications in the α-syn-coding gene (duplication or triplication) and, therefore, may represent a suitable in vivo model for investigating synucleinopathies [167,168]. However, they do not produce Lewy body inclusions, a feature that is considered a pathological hallmark of PD in humans, representing, therefore, the main limitation of these two invertebrate models [169].

Kerr and colleagues used an inducible *Drosophila melanogaster* model of AD and confirmed the ability of Aβ42 to inhibit Nrf2 activity in neurons. In addition, loss-of-function mutations in Keap1 significantly protected against Aβ1-42-induced toxicity [170]. TDP-43 (a highly conserved nuclear RNA/DNA-binding protein involved in the regulation of RNA processing [171]) is involved in the pathology of familial and sporadic ALS, and distinct TDP-43-mediated ALS models, including *Drosophila melanogaster* and zebrafish, exhibit defective neuromuscular junctions, dysfunction of locomotor function, and motor neuron defects. Cha et al. analyzed the protective mechanism of glutathione S-transferase omega 2 (GstO2) on the neurotoxicity of human TDP-43 protein (hTDP-43) by measuring the intracellular ROS levels in the *Drosophila melanogaster* brain. Expression of hTDP-43 induced motor neuronal toxicity and significantly increased ROS levels in fly neurons, which could suggest the activation of ferroptosis. In addition, GstO2 was capable of reducing both the degenerative and defective phenotypes, and markedly decreased intracellular ROS generation, thus suggesting GstO2 as a key regulator of hTDP-43-related ALS pathogenesis [172].

In *Drosophila melanogaster*, frataxin deficiency (a hallmark of FRDA) induces iron hypersensitivity and a reduced life span. In addition, after 7 days of iron treatment, frataxin-deficient flies displayed a significant reduction in the activity of aconitase and complex II [173], leading to mitochondria damage. This indicates the usefulness of the *Drosophila melanogaster* model to study ferroptosis mechanisms in FRDA and other NDs.

The *C. elegans* model has been increasingly used to study several NDs, as it can be genetically modified to express the pathophysiological alterations of such pathologies. Some studies have been using this animal model to investigate ferroptosis in neurodegeneration. Jenkins and colleagues showed that acute GSH depletion with diethyl maleate (DEM; at concentrations equal or above 1 mM) resulted in a significant increase of MDA and 4-HNE levels, and induced death of 4-day-old adult worms. Nonetheless, DEM toxicity was reversed by the ferroptosis inhibitor Lip-1 (200 μM), suggesting the involvement of ferroptosis in the phenotype [174]. Vázquez-Manrique and colleagues showed that the shortening of the frataxin gene in *C. elegans* reduced lifespan, and worms have increased sensitivity to oxidative stress, which might explain the reduction of longevity [175]. Overall, these data suggest a great potential for *C. elegans* to study ferroptosis in NDs, particularly in FRDA.

Peres et al. have shown the ability of dihomo-gamma-linolenic acid (DGLA) to trigger ferroptosis in *C. elegans* germ cells and an increased sensitivity to ferroptosis in ether lipid-deficient mutant strains [176]. This highlighted a protective role for ether lipids against DGLA-induced ferroptosis. Following these observations, they further showed a lower sensitivity of both ether-lipid deficient mutants and control animals to ferroptosis by changing the abundance of monounsaturated fatty acids, saturated fats, and PUFA. In particular, the sensitivity to ferroptosis was reduced in mutant strains unable to synthesize DGLA [strain carrying a gain-of-function mutation in the transcriptional mediator MDT-15 (mediator of RNA polymerase II transcription subunit 15, a transcriptional co-activator involved in several processes, including fatty acid desaturation and stress response)] and by dietary supplementation with monounsaturated fatty acids. Overall, these data demonstrated that dietary lipids may greatly influence germ cells’ sensitivity to ferroptosis. It also indicated an important role for both dietary and endogenous monounsaturated fatty acids in protecting from/preventing ferroptosis [177].

Zebrafish represent a powerful experimental tool for NDs research, since the molecular mechanisms underlying these pathologies are conserved. In addition, zebrafish presents numerous methodological advantages over other models (Appendix A) [178]. SOD1 is considered a remarkably powerful antioxidant enzyme involved in cell protection against the damaging effects of superoxide radicals, and which can be mutated (with gain or loss of function) [179]. As seen in other models, mutations in SOD1 induce oxidative stress in embryonic zebrafish (reduction of embryo survival upon incubation with 5 mM hydrogen peroxide) [180]. Since oxidative stress is one of the mechanisms that occur in ferroptosis, this supports the usefulness of zebrafish to study this type of cell death in ALS and other NDs.

#### 4.2.3. Other Animal Models

The rodent models have provided important insights into the pathogenesis of NDs. However, most of the rodent models cannot fully mimic the pathophysiology and symptoms of NDs, as differences exist between rodent and human brains (anatomic structure, physiology, and neuropathology). This requires, therefore, the use of more complex animal models, such as rabbits, pigs, dogs, and monkeys [181,182].

Nonetheless, these complex models have several limitations, including low efficiency of homologous recombination, long-life cycle, and lack of embryonic stem cells (ESCs) for in vitro genome editing by the developed nuclease-mediated genome editing technology CRISPR/Cas9 [182]. The main advantages and disadvantages of these models are represented in Figure 5 and listed in Appendix A.

A study developed in 2003 analyzed oxidative stress markers for lipid peroxidation (4-HNE) and glycoxidation [lipofuscin-like pigment (LFP)] in a canine model of AD. An accumulation of 4-HNE was observed in neurons and macrophages, the same as for lipofuscin pigment, and both increased with age, as expected in an AD model. Thus, this animal model recapitulates the increased oxidative stress and lipid peroxidation seen in AD [183]. Another study analyzed the levels of LFP, protein carbonyls, and vitamin E in the brain of a canine model of senile dementia (ccSDAT) like AD. These animals showed an intensive production of free radicals in the brain, as well as significantly higher levels of LFP and protein carbonylation, as compared to age-matched controls. By contrast, vitamin E concentration was significantly decreased in ccSDAT animals. These results indicated that canine models recapitulate pathological features of AD. In addition, as uncontrolled lipid peroxidation represents a hallmark of ferroptosis, these studies also suggest that canine models might be useful to explore the involvement of ferroptosis in NDs [184].

In another study, New Zealand white rabbits were used to study PD, where the effects of MPP^+^ in the SN were explored by directly administrating the neurotoxin into the cerebrospinal fluid. MPP^+^ reduced the tyrosine hydroxylase-positive cell population in the SN, demonstrating that the rabbit is a sensitive model to MPP^+^. In addition, it also highlights the potential of this model in NDs research [185].

Ferroptosis involvement in NDs has not been explored in these animal models. However, since they recapitulate different pathological features of NDs, such as oxidative stress and lipid peroxidation, which are active players in ferroptosis, it suggests their promising potential to study ferroptosis in neurodegeneration.

## 5. Conclusions

A new type of programmed cell death, called ferroptosis, was identified by Dixon in 2012. Ferroptosis is characterized by GSH depletion, decreased GPX4 activity, and increased generation of ROS through the Fenton reaction, lipid peroxidation, and iron accumulation. Over the past decade, a significant amount of attention has been paid to ferroptosis in NDs, with the goal of identifying novel therapeutic strategies for such conditions.

Given the impact of ferroptosis in the pathogenesis of several diseases, numerous preclinical and clinical trials have been performed and are continually being designed and implemented to evaluate the effectiveness of different ferroptosis inhibitors in the treatment and prevention of NDs. In addition, the brain is particularly vulnerable to lipid peroxidation, as it has the highest levels of PUFAs in the human body, which represent well-known lipid peroxide precursors. In fact, a close correlation between GSH depletion, lipid peroxidation, and NDs is well documented.

Although ferroptosis involves several physiological and pathological processes, the use of distinct and appropriate models is imperative to achieve a reliable transition from basic research to clinical studies. With special attention to NDs, several in vitro or in vivo models have been used to recapitulate the diseases’ phenotypes. The use of such models is useful to explore in detail the mechanistic aspects of ferroptosis, their involvement in NDs, and to discover potential inhibitors of these processes. Such knowledge may lead to the development of potentially new disease-modifying drugs capable of stopping or delaying disease progression.

On the other hand, it becomes extremely important to adapt these in vitro and in vivo models to NDs, mimicking the different pathological mechanisms, including ferroptosis. This is particularly relevant as ferroptosis is not yet fully standardized or clarified, possibly because it represents a recently described type of cell death. Therefore, the study of ferroptosis in distinct models represents a strategy for future research, as it allows for obtaining higher confidence in the role of ferroptosis in the pathogenesis of distinct NDs.

## Figures and Tables

**Figure 1 pharmaceutics-15-01369-f001:**
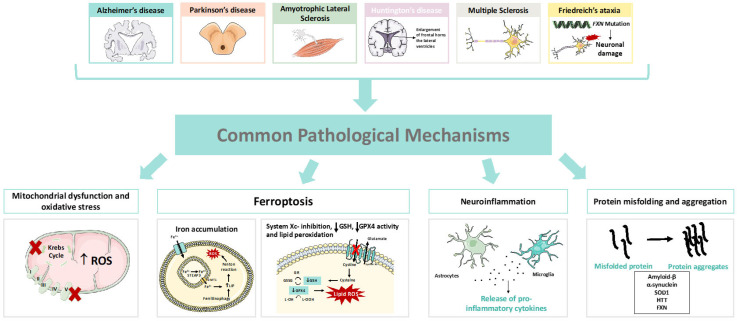
Pathological mechanisms common to several neurodegenerative diseases (Alzheimer’s disease, Parkinson’s disease, amyotrophic lateral sclerosis, Huntington’s disease, multiple sclerosis, and Friedreich’s ataxia).

**Figure 2 pharmaceutics-15-01369-f002:**
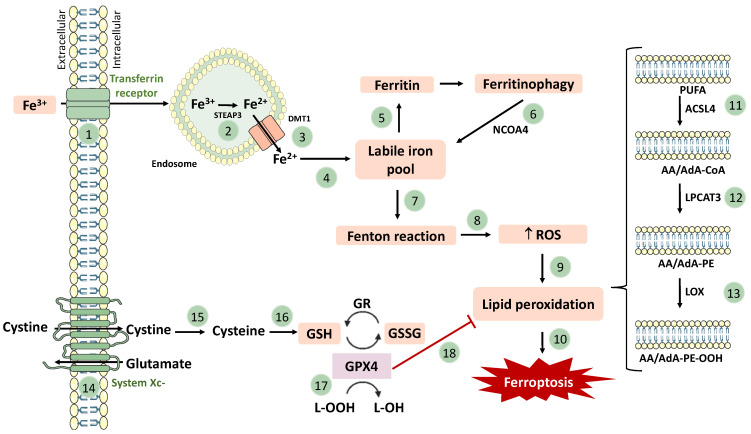
Metabolic pathways that contribute to ferroptosis. Ferroptosis is a type of regulated cell death characterized by multiple pathological mechanisms, including iron overaccumulation. Fe^3+^ connects with transferrin, and (1) the complex iron-transferrin enters the cell through the transferrin receptor. Within the endosome, (2) Fe^3+^ is reduced to Fe^2+^ through the activity of a six-transmembrane epithelial antigen of prostate 3 (STEAP3). Lastly, (3) Fe^2+^ is translocated across the endosomal membrane back to the cytosol by the divalent metal transporter 1 (DMT1), forming the (4) labile iron pool. Alternatively, (5) the excess of iron is stored bound to ferritin, representing a redox-inactive form of iron that protects cells and tissues from iron-induced damage. Ferritin degradation, namely by (6) ferritinophagy, which is mediated by nuclear receptor coactivator 4 (NCOA4), increases the labile iron pool, contributing to ferroptosis. The (7) excess of iron drives the Fenton reaction, with (8) generation of high amounts of ROS, which contribute to (9) lipid peroxidation and to (10) ferroptosis. PUFAs are the preferential substrates for lipid peroxidation due to their structure. This process is initiated by the formation of (11) arachidonic acid (AA)/adrenic acid (AdA)-Coenzyme A (Co-A) by Acyl-CoA Synthetase Long Chain Family Member 4 (ACSL4); (12) Lysophosphatidylcholine Acyltransferase 3 (LPCAT3) then conjugates AA/AdA-CoA with phosphatidylethanolamine (PE) to form AA/AdA-PE. The enzymatic pathway, which is dependent on (13) lipoxygenases (LOX), promotes the peroxidation of AA/AdA-PE to AA/AdA-PE-OOH. (14) System Xc- exchanges extracellular cystine with intracellular glutamate. (15) Cystine is then intracellularly reduced to cysteine, which is used for the (16) biosynthesis of reduced glutathione (GSH). Furthermore, cytotoxic lipid hydroperoxides (L-OOH) are reduced to the corresponding alcohols (L-OH) by (17) glutathione peroxidase 4 (GPX4), while oxidizing GSH into glutathione disulphide (GSSG), thus limiting ferroptosis by (18) inhibiting lipid peroxidation. An inhibition of system Xc- activity decreases cystine import, affecting GSH biosynthesis and consequently decreasing GPX4 activity. The cell’s antioxidant capacity will remarkably decrease, promoting the accumulation of lipid ROS and, ultimately, ferroptosis.

**Figure 3 pharmaceutics-15-01369-f003:**
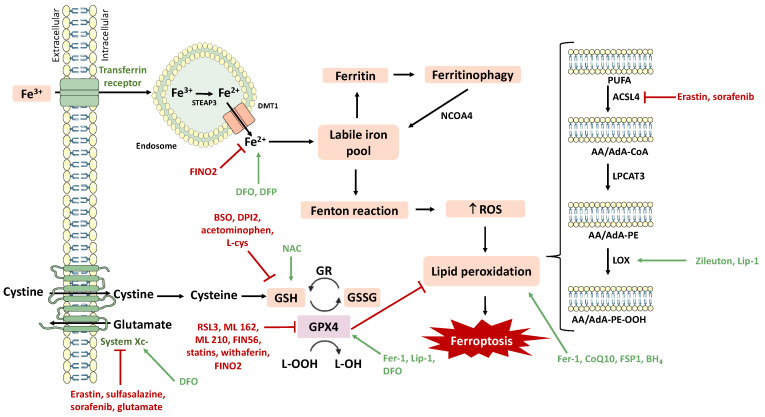
Ferroptosis inducers and inhibitors and their corresponding targets. Ferroptosis inducers are indicated in red and ferroptosis inhibitors are indicated in green.

**Figure 4 pharmaceutics-15-01369-f004:**
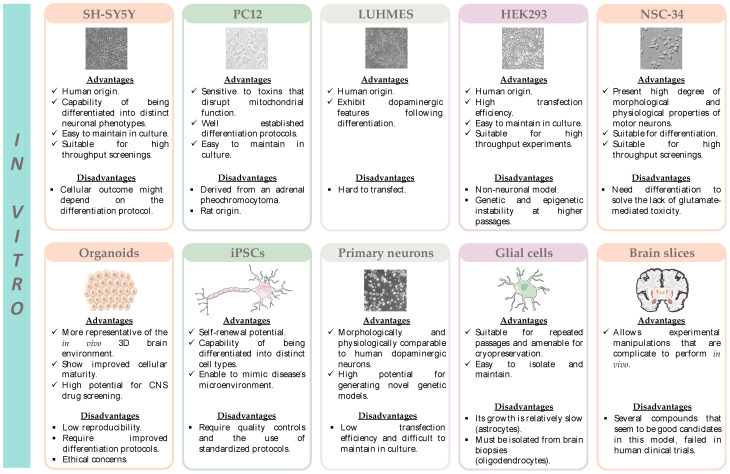
Main advantages and disadvantages of different in vitro models used in the study of distinct neurodegenerative diseases.

**Figure 5 pharmaceutics-15-01369-f005:**
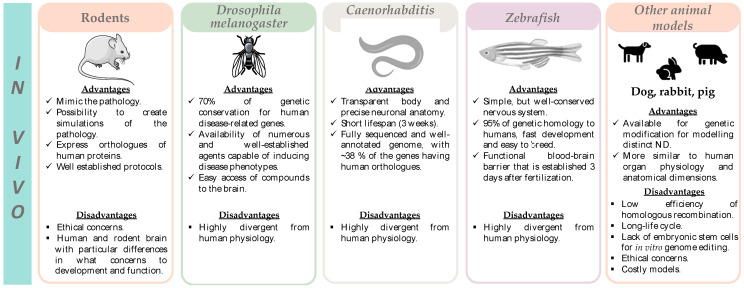
Main advantages and disadvantages of different in vivo models used in the study of distinct neurodegenerative diseases.

**Table 2 pharmaceutics-15-01369-t002:** Compilation of different studies using distinct in vivo models for the evaluation of ferroptosis’ involvement in neurodegenerative diseases.

Model		Main Conclusions	Reference
**Rodents**	**Model**: APP knockout in mouse models (APP-/-) Ferroptosis inhibitor: **DFP** (50 mg/kg/day in drinking water for 3 months)	APP-/- mouse model presented an iron elevation in the CNS, but DFP ameliorated the SN neuronal loss.	[152]
	**Model**: 3 and 6 month old Nrf2+/+ and Nrf2-/- mice that overexpress human α-syn (hα-Syn+:Nrf2+/+ and hα-Syn+:Nrf2-/-)	Nrf2 is a critical anti-ferroptotic mediator of neuronal survival, and the vicious cycle of α-syn overexpression and Nrf2 suppression leads to an enhanced neuronal ferroptotic cell death.	[153]
	**Model**: triple transgenic mice (3xTg mice, harboring the APP Swedish mutation (KM670/671NL), the M146V human mutant PS1, and the Tau P301L) Ferroptosis inhibitor: **Zileuton** (200 mg/L of drinking water for three months)	Aβ1-42 levels were significantly lower in the brain of the zileuton-treated animals; besides, a significant reduction in the amount of Aβ peptides deposited in the brain of zileuton-treated-animals was also observed.	[156]
	**Model**: APPswe/PS1E9 transgenic mice Ferroptosis inhibitor: **Eriodictyol** (50 mg/kg, i.p. for 3 months)	Eriodictyol reduced Aβ aggregation, decreased the levels of Aβ peptide and *p*-tau, ferrous iron, total iron, TfR and FTH in the cortex and hippocampus and decreased MDA content, and also an increased GPX4 expression levels.	[157]
	**Model**: Aβ1-42 AD mice Ferroptosis inhibitor: **Salidroside** (50 mg/kg/day)	Salidroside attenuated mitochondrial changes and alleviated the decrease in GPX4 levels caused by Aβ1-42 in the hippocampus and cortex.	[158]
	**Model**: Cuprizone model of MS	Increased NCOA4, TfR1 and COX2 activity and augmented 4-HNE levels. GPX4 levels were also significantly reduced, as well as the expression of system Xc-.	[86]
	**Model**: Rat model of Temporal lobe epilepsy induced by LiCl-Pilo Ferroptosis inhibitor: **klotho** (overexpressed using an adeno-associated viral vector)	Klotho was mainly located in the neurons, rather than in the astrocytes. FPN levels were increase in animals exposed to klotho, and the opposite occurred in DMT-1 levels. Also, the levels of GSH and GPX4 in the hippocampus were significantly elevated, while ROS levels were suppressed.	[165]
	**Model**: transgenic mice expressing 5 familial AD mutation (5xFAD mice) Ferroptosis inhibitor: **OABL** (20 mg/kg/day i.p.)	OABL inhibited mitochondrial changes, increased GSH levels and reduced MDA content, and decreased the levels of Aβ1-42 and phosphorylated Tau observed in 5xFAD mice.	[116]
	**Model**: Delayed neurocognitive recovery aged mice Ferroptosis inhibitor: **MSCs-Exo** and **Fer-1** (1.5 mg/kg)	MSCs-Exo inhibited ferroptosis in dNCR aged mice, as same as Fer-1 (positive control).	[166]
	**Model**: Rat model of PD (mice exposed to 25 μg 6-OHDA in the medial forebrain bundle)	6-OHDA decreased, in vivo, the levels of TH and GPX4, decreased the GSH/GSSG ratio and increased MDA levels, suggesting increased lipid peroxidation.	[115]
** *Drosophila melanogaster* **	**Model**: *Drosophila melanogaster* model of AD	Aβ42 was able to inhibit the activity of Nrf2 in neurons and loss-of-function mutations in Keap1 significantly protected against Ab42-induced toxicity.	[170]
	**Model**: *Drosophila melanogaster* expressing hTDP-43-expressing Ferroptosis inhibitor: **GstO2**	GstO2 was proposed as a key regulator of hTDP-43-related ALS pathogenesis and highlighted as a potential target in the treatment of ALS.	[172]
	**Model**: *Drosophila melanogaster* with frataxin deficiency	This deficiency induced iron hypersensitivity, a reduction life span and a reduction in the activity of aconitase and complex II, leading to mitochondria damage.	[173]
** *Caenorhabditis elegans* **	Ferroptosis inducer: DEM (1 mM) Ferroptosis inhibitor: **Lip-1** (200 μM)	DEM induced death of 4-day old adult worms, but the ferroptosis inhibitor reversed this death. DEM promoted an acute GSH depletion that triggered a marked increase in the levels of MDA and 4-HNE, which was ameliorated by Lip-1.	[174]
	**Model**: *C. elegans* with a reduction of the frataxin gene	Shortening the frataxin gene increased the animal sensitivity to oxidative stress.	[175]
	Ferroptosis inducer: DGLA **Model**: Ether lipid-deficient mutant strains	Dietary lipids may greatly influence germ cells’ sensitivity to ferroptosis.	[176]
**Zebrafish**	**Model**: Embryonic zebrafish Compound: **Hydrogen peroxide** (5 mM)	Mutant *SOD1* significantly induced oxidative stress (which was demonstrated by a reduction in survival of embryos upon incubation with hydrogen peroxide), potentially promoting ferroptosis (as oxidative stress induces the depletion of antioxidant defenses such as GSH, being GSH depletion a hallmark of ferroptosis).	[180]

α-syn: a-synuclein; 4-HNE: 4-hydroxy-2-nonenal; 6-OHDA: 6-hydroxydopamine; Aβ: Amyloid-β; ALS: Amyotrophic lateral sclerosis; CNS: Central nervous system; COX2: Cyclooxygenase 2; DEM: Diethyl maleate; DFP: Deferiprone; DGLA: Dihomo-gamma-linolenic acid; DMT-1: Divalent metal transporter 1; dNCR: Delayed neurocognitive recovery; Fer-1: Ferrostatin-1; FPN: Ferroportin; FTH: Ferritin heavy chain; GPX4: Glutathione peroxidase 4; GSH: Glutathione; GSSG: Glutathione disulfide; GstO2: Glutathione S-transferase omega 2; Keap1: Kelch-like ECH-related protein 1; Lip-1: Liproxstatin-1; MDA: Malondialdehyde; MS: Multiple sclerosis; MSCs-Exo: Mesenchymal stem cell-derived exosomes; NCOA4: Nuclear receptor coactivator 4; Nrf2: Nuclear factor erythroid 2–related factor 2; OABL: 1,6-O,O-diacetyl-britannilactone; ROS: Reactive oxygen species; SOD1: Superoxide dismutase 1; TfR: Transferrin receptor; TH: Tyrosine hydroxylase; TLE: Temporal lobe epilepsy.

## Data Availability

Not applicable.

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
