# Peer review of "Research Models to Study Ferroptosis’s Impact in Neurodegenerative Diseases"

_pharmaceutics, 2023, doi:10.3390/pharmaceutics15051369_

Round 1
Reviewer 1 Report
In this manuscript, authors Research models to study ferroptosis’s impact in neurodegenerative diseases . In my opinion, some issues should be further address and I hope following comments could be helpful for improving their paper.
- In the introduction, the background about ferroptosis is little, the authors should enrich this part by citing some recent literatures and emphasize the necessity of the treatment of neurodegenerative diseases
- This manuscript is well organized but lack of specific comparative analysis. What are the advantages of ferroptosis compared with other therapeutic models? Authors need to explain it and also include it in the manuscript.
- The caption for figure 1 and 2is too lengthy, kindly revised it to more precise
- Good quality of figures are very important for review paper. Kindly add at least 4-6 more figures. It will be better to take it from recent literatures and correlate it with your ideas.
- Authors just explains two in vivo models, kindly also add more in vivo models and elaborate it further in detail.
- In conclusions kindly add some future perspectives, the author should consider giving some methodological design about how to improve the performance of such materials.
- Kindly add challenges in this review paper
Author Response
Reviewer #1:
In this manuscript, authors Research models to study ferroptosis’s impact in neurodegenerative diseases. In my opinion, some issues should be further address and I hope following comments could be helpful for improving their paper.
Response to reviewer: We acknowledge the reviewer’s concerns. According to the diverse suggestions raised by reviewer 1, several changes were made in the revised version of the manuscript, which greatly improved its overall quality. We hope that all the performed improvements can fulfill the Reviewer’s expectations.
- In the introduction, the background about ferroptosis is little, the authors should enrich this part by citing some recent literatures and emphasize the necessity of the treatment of neurodegenerative diseases.
Response to reviewer: We acknowledge the reviewer’s suggestions and a deeper explanation of ferroptosis involvement in distinct neurodegenerative diseases was added to the revised version of the manuscript. Also, a new figure illustrating the common pathophysiological mechanisms underlying such diseases, including ferroptosis, was added to the revised manuscript. Additionally, recent literature references, including an extensive overview on ferroptosis and neurodegeneration recently published by our group, were added to the revised manuscript.
- This manuscript is well organized but lack of specific comparative analysis. What are the advantages of ferroptosis compared with other therapeutic models? Authors need to explain it and also include it in the manuscript.
Response to reviewer: We acknowledge the reviewer’s suggestions. The main aim of this manuscript is to explore the potential in vitro and in vivo models that can be used to successfully clarify ferroptosis impact in distinct neurodegenerative diseases, and to better elucidate the potential therapeutic application of ferroptosis inhibitors in the treatment of such diseases. The explored neurodegenerative diseases present a multifactorial nature, with several pathophysiological mechanisms coursing together, many of them common among these diseases. Furthermore, most of the in vitro and in vivo models explored in the manuscript can be used not only to study ferroptosis but also other pathophysiological mechanism involved in such diseases.
- The caption for figure 1 and 2 is too lengthy, kindly revised it to more precise.
Response to reviewer: We acknowledge the reviewer’s suggestions and the captions of both figures 1 and 2 (now figure 2 and 3 of the revised manuscript) were simplified in the revised version of the manuscript. In addition, additional figures were added to the manuscript with a summarized description of their contents in the respective figure legend.
- Good quality of figures are very important for review paper. Kindly add at least 4-6 more figures. It will be better to take it from recent literatures and correlate it with your ideas.
Response to reviewer: We acknowledge the reviewer’s suggestions, which greatly improved the overall quality of the manuscript. Accordingly, three new figures were added to the revised version of the manuscript, one focusing the common pathophysiological mechanisms underlying the neurodegenerative diseases mentioned in the manuscript (Figure 1), and two other figures focusing on the advantages and disadvantages of the selected in vitro and in vivo models (Figures 4 and 5). The corresponding tables where this information was initially compiled were transferred to the supplementary material (Tables S1 and S2 of the revised manuscript). This way, the manuscript becomes more appealing to the readers and easy to follow.
- Authors just explains two in vivo models, kindly also add more in vivo models and elaborate it further in detail.
Response to reviewer: We acknowledge the reviewer’s suggestions. The manuscript focused 4 in vivo models (rodents, Drosophila melanogaster, Caenorhabditis elegans and Zebrafish). This is because, to the best of our knowledge, no other animal models have been used to explore ferroptosis. However, to accomplish with the reviewer’s suggestions, in the revised version of the manuscript we incorporated 2 additional in vivo models, as 3 studies showed increased lipid peroxidation in canine and rabbit models of neurodegenerative diseases. Therefore, considering that lipid peroxidation represents a hallmark of ferroptosis, this suggest that these models might be useful to explore ferroptosis in neurodegeneration.
- In conclusions kindly add some future perspectives, the author should consider giving some methodological design about how to improve the performance of such materials.
Response to reviewer: We acknowledge the reviewer’s suggestion and some future perspectives were accordingly added to the conclusion of the revised manuscript. In particular, the following sentence was added:
” On the other hand, it becomes extremely important to adapt these in vitro and in vivo models to neurodegenerative diseases, mimicking the different pathological mechanisms, including ferroptosis. This is particularly relevant as ferroptosis is not yet fully standardized or clarified, possibly because it represents a recently described type of cell death. Therefore, the study of ferroptosis in distinct models represents a strategy for future research, as it allows to obtain a higher confidence on the role of ferroptosis in the pathogenesis of distinct ND”.
- Kindly add challenges in this review paper
Response to reviewer: We acknowledge the reviewer’s suggestions, which were all accomplished in the revised manuscript and greatly improved its overall quality.

Reviewer 2 Report
This is an excellent review focused on the relationship between the ferroptosis and neurodegenerative diseases. Overall, the paper is well written. Nevertheless, there are several issues that should be addressed.
(1)Line180: “(ACSL4), lipoxygenase (LOX) and Lysophosphatidylcholine Acyltransferase 3 180
(LPCAT3) (Figure 1) “. ACSL4, LOX, LPCAT3 should be included in Figure 1.
(2)Line199: “Sulfasalazine, Sorafenib and Glutamate” should be “sulfasalazine, sorafenib and glutamate” like line 263.
(3)“System Xc” or “system Xc” need to be uniform, it is recommended to use lowercase for the first letter like line 264.
(4)Figure 1 and Figure 2: GSH to GSSG arrow is missing.
(5)Some prospectives should be added in the section of conclusion.
Author Response
Reviewer comments:
Reviewer #2:
This is an excellent review focused on the relationship between the ferroptosis and neurodegenerative diseases. Overall, the paper is well written. Nevertheless, there are several issues that should be addressed.
Response to reviewer: We greatly acknowledge the reviewer’s comment on the overall quality of the manuscript, as well as the raised suggestions, which were all accomplished in the revised manuscript and greatly improved its overall quality.
(1)Line180: “(ACSL4), lipoxygenase (LOX) and Lysophosphatidylcholine Acyltransferase 3 180 (LPCAT3) (Figure 1) “. ACSL4, LOX, LPCAT3 should be included in Figure 1.
Response to reviewer: We greatly acknowledge the reviewer’s suggestion and ACSL4, LOX, LPCAT3 were included in Figure 2 and Figure 3 of the revised manuscript (former Figures 1 and 2).
(2)Line199: “Sulfasalazine, Sorafenib and Glutamate” should be “sulfasalazine, sorafenib and glutamate” like line 263.
Response to reviewer: We greatly acknowledge the reviewer’s suggestions, and these corrections were introduced in the revised version of the manuscript.
(3)“System Xc” or “system Xc” need to be uniform, it is recommended to use lowercase for the first letter like line 264.
Response to reviewer: We greatly acknowledge the reviewer’s suggestions and an uniformization was made throughout the revised version of the manuscript.
(4)Figure 1 and Figure 2: GSH to GSSG arrow is missing.
Response to reviewer: We greatly acknowledge the reviewer’s suggestions and the Figures were corrected in the revised version of the manuscript (now Figures 2 and 3 of the revised manuscript).
(5) Some prospectives should be added in the section of conclusion.
Response to reviewer: We acknowledge the reviewer’s suggestion and some prospectives were accordingly added to the conclusion of the revised manuscript. In particular, the following sentence was added:
” On the other hand, it becomes extremely important to adapt these in vitro and in vivo models to neurodegenerative diseases, mimicking the different pathological mechanisms, including ferroptosis. This is particularly relevant as ferroptosis is not yet fully standardized or clarified, possibly because it represents a recently described type of cell death. Therefore, the study of ferroptosis in distinct models represents a strategy for future research, as it allows to obtain a higher confidence on the role of ferroptosis in the pathogenesis of distinct ND”.

Reviewer 3 Report
The manuscript from Inês Costa and colleagues reviewed the main in vitro and in vivo models that can be used to evaluate ferroptosis in the most prevalent neurodegenerative diseases and to explore potential new drug targets and novel drug candidates for effective disease-modifying therapies.
The update was done very carefully. I think that it fully exhausts the issues raised by the authors. I only have some minor concerns as below:
1, The introduction section needs to be more concise when introducing the background of various neurodegenerations. They should describe the connection between ferroptosis and neurodegenerations in more detail.
2, It will be appreciated if they can add more in vivo models and elaborate it further in detail.
Author Response
Reviewer comments:
Reviewer #3:
The manuscript from Inês Costa and colleagues reviewed the main in vitro and in vivo models that can be used to evaluate ferroptosis in the most prevalent neurodegenerative diseases and to explore potential new drug targets and novel drug candidates for effective disease-modifying therapies.
The update was done very carefully. I think that it fully exhausts the issues raised by the authors. I only have some minor concerns as below:
Response to reviewer: We greatly acknowledge the reviewer’s feedback on the overall quality of the manuscript. Additionally, several changes were performed in the revised version of the manuscript, which further contributed for its overall improvement.
1, The introduction section needs to be more concise when introducing the background of various neurodegenerations. They should describe the connection between ferroptosis and neurodegenerations in more detail.
Response to reviewer: We acknowledge the reviewer’s suggestion and a deeper explanation on the ferroptosis involvement in the neurodegenerative diseases explored was introduced in the revised version of the manuscript.
2, It will be appreciated if they can add more in vivo models and elaborate it further in detail.
Response to reviewer: We greatly acknowledge the reviewer’s suggestions. Thus, in the revised version of the manuscript we incorporated 2 additional in vivo models, as 3 studies show increased lipid peroxidation in canine and rabbit models of neurodegenerative diseases. Therefore, considering that lipid peroxidation represents a hallmark of ferroptosis, this suggest that these models might be useful to explore ferroptosis in neurodegeneration.

Round 2
Reviewer 1 Report
Accpeted